# All-fiber tribo-ferroelectric synergistic electronics with high thermal-moisture stability and comfortability

Weifeng Yang[1,4], Wei Gong[1,4], Chengyi Hou[1]*, Yun Su[2], Yinben Guo[1], Wei Zhang[1], Yaogang Li[3], Qinghong Zhang ⬤ [3]* & Hongzhi Wang[1]*

Developing fabric-based electronics with good wearability is undoubtedly an urgent demand for wearable technologies. Although the state-of-the-art fabric-based wearable devices have shown unique advantages in the field of e-textiles, further efforts should be made before achieving "electronic clothing" due to the hard challenge of optimally unifying both promising electrical performance and comfortability in single device. Here, we report an all-fiber tribo-ferroelectric synergistic e-textile with outstanding thermal-moisture comfortability. Owing to a tribo-ferroelectric synergistic effect introduced by ferroelectric polymer nanofibers, the maximum peak power density of the e-textile reaches $5.2\,\mathrm{W\,m^{-2}}$ under low frequency motion, which is 7 times that of the state-of-the-art breathable triboelectric textiles. Electronic nanofiber materials form hierarchical networks in the e-textile hence lead to moisture wicking, which contributes to outstanding thermal-moisture comfortability of the e-textile. The all-fiber electronics is reliable in complicated real-life situation. Therefore, it is an idea prototypical example for electronic clothing.

[1] State Key Laboratory for Modification of Chemical Fibers and Polymer Materials, College of Materials Science and Engineering, Donghua University, Shanghai 201620, PR China. [2] College of Fashion and Design, Donghua University, Shanghai 200051, PR China. [3] Engineering Research Center of Advanced Glasses Manufacturing Technology, Ministry of Education, Donghua University, Shanghai 201620, PR China. [4] These authors contributed equally: Weifeng Yang, Wei Gong *email: hcy@dhu.edu.cn; zhangqh@dhu.edu.cn; wanghz@dhu.edu.cn

Along with the springing up vigorous emergence of wearable electronics, closely related issues of endurance and wearability have become an important bottleneck restricting its development. Electronic textiles (e-textiles), as a new generation wearable electronics, show promising wearability but still face many difficulties in terms of safety, practicality and comfortability[1,2]. The shortcoming is the inadequate wearable energy technology. It remains a challenge to optimally unify appropriate electrical properties and wearability in single energy device. For instance, flexible rechargeable batteries[3] and supercapacitors[4] require repeated charging, while the safety and stability are affected during frequent bending and deformation. Photovoltaic cells[5,6] suffer similar issues of mechanical stability, and also are highly dependent on working environment. Thermoelectric devices[7,8] made of low-dimensional materials are flexible but have very limited power density. Moreover, above technologies are hardly applied in fiber- and fabric-like devices.

As a comparison, triboelectric technology[9,10] has shown great potential in the field of e-textiles because of its high efficiency in low-frequency and random mechanical energy collection, as well as its various possibilities of material and device design including nanofibers and fabrics. However, the power density of current triboelectric textiles is generally low. Though coupling polarized inorganic piezoelectric/ferroelectric nanoparticles to modify charge trapping capability can effectively increase electrical output[11–13], the harsh polarization processes and dense film morphology of materials severely limit its utility in textiles. Moreover, few studies have considered the thermal-moisture stability and comfortability of e-textiles, which are crucial indicators for evaluating wearability[14,15]. Especially for triboelectric textiles, electrical performance decay in changeable thermal-moisture conditions on body surface is a major issue that still need to be addressed[16,17].

To develop e-textiles that simultaneously meet rigorous electrical and wearable performance requirements, we set an "all-fiber" principle: First, we employ only nanofiber materials in producing tribo-ferroelectric synergistic effect so as to improve output of triboelectric devices while remain good flexibility. Second, all function layers of the device are built by fiber networks to guarantee breathability and moisture permeability. Third, nanofiber network is constructed into hierarchical structure for realizing moisture wicking function of the e-textile. Owing to above designs, the e-textile has a high electrical output as well as outstanding thermal-moisture stability and comfortability. The maximum peak power density of the e-textile reaches 5.2 W m$^{-2}$ under low frequency (~2.5 Hz) motion, which is 7 times that of the state-of-the-art breathable triboelectric textiles. We also demonstrate several real-life applications of the all-fiber electronics readily as clothing, including powering the LCD, digital electroluminescent lattices and electronic watch, and monitoring human motions. It provides a practical and propagable route for the development of next-generation electronic clothing.

## Results

### The structure of the e-textile and the proposal of tribo-ferroelectric synergy model.
Figure 1a depicts the all-fiber tribo-ferroelectric synergistic e-textile, which consists of four function fabric layers, including two nanofiber nonwovens poly(vinylidene fluoride-trifluoroethylene) (P(VDF-TrFE)) and polyamide 6 (PA6) with opposite tribo-polarity for contact electrification, nickel–copper (Ni–Cu) fabric electrode for charge induction, and the moisture-wicking fabric for directional water transport and rapid evaporation. The P(VDF-TrFE) nanofibers also act as a polymer ferroelectricity (defined as inner/outer ferroelectric

layers) for constructing tribo-ferroelectric synergistic enhancement effect. Electrospinning was adopted to induce rich ferroelectric β-phase as well as the steering polarization of CF$_2$ dipoles (defined as primary polarization)[18,19] in P(VDF-TrFE) nanofibers (Supplementary Figs. 1 and 2).

Figure 1b illustrates the ferroelectric charge and discharge mechanism of the all-fiber electronics. Regulating the electrostatic energy stored inside of P(VDF-TrFE) ferroelectricity in the form of electric displacement polarization to promote charge transfer between two tribo-polarity materials. From the perspective of ferroelectric energy storage, when applied electric field changes continuously, the P(VDF-TrFE) nanofibers act as an equivalent capacitor for reciprocating charging and discharging[20]. The energy density stored in P(VDF-TrFE) nanofibers can be expressed as

$$U_s = U - U_r = \int_0^{P_m} E dP - \int_{P_r}^{P_m} E dP \qquad (1)$$

where $U$ is the energy for charging ferroelectricity during the increase of electric field, $U_s$ and $U_r$ are the energy stored in and released from ferroelectricity during the change of applied electric field ($E$), $P_m$ and $P_r$ are the maximum and remnant polarization intensity, respectively. We used the Sawyer–Tower method[21] (Supplementary Fig. 3) to test the hysteresis loop of P(VDF-TrFE) nanofiber nonwovens (Fig. 1c). The maximum and residual polarization intensity are about 2.03 and 1.05 μC cm$^{-2}$ under the electric field intensity of 1000 kV cm$^{-1}$, respectively. When applied electric field increases in positive direction, the CF$_2$ dipoles continuously turn and increase the polarization of ferroelectric phase rapidly until saturation (state 1 to 2). If electric field is reduced at this time, some dipoles will relax and the polarization will gradually decrease (state 2 to 3). When the electric field decreases to 0, some dipoles remain polarized and exhibit a residual polarization intensity.

Figure 1d is a schematic diagram of the tribo-ferroelectric synergistic enhancement model in dual-ferroelectric polarized (DFP) e-textile. Owing to contact electrification and electrostatic induction, the electrons transfer from PA6 to P(VDF-TrFE) nanofibers while charges are generated at fabric electrodes. Therefore, an electrostatic field forms between dielectric and conductive materials which makes a secondary polarization on P(VDF-TrFE) nanofiber ferroelectricity corresponding to the state 1, 2, and 3 processes of the hysteresis loop (Fig. 1c). Ferroelectricity is continuously charged and discharged in the cycle of contact and separation. The electrostatic energy stored in ferroelectricity will promote charge transfer and improve performance of the e-textile, while the polarization of ferroelectricity can be promoted as a result of the increase of charge density. According to the ferroelectric capacitance model theory[22,23], the energy required for dipoles rotating to equilibrium position can be expressed as:

$$\Delta U = \mu \left( E_D - \frac{d}{D} \cdot E_d + E_p \right)(\cos \theta + 1) \qquad (2)$$

where $E_0$, $E_D$, $E_d$, $E_p$, $E_e$ are the electric field intensity between dielectric and conductive layer, inside the ferroelectric layer, inside the PA6 layer, generated by the other dipoles and acting on dipoles effectively, respectively. $D$ and $d$ are the thickness of ferroelectric layer and PA6, $\mu$ is dipole moment, $\theta$ is the angle between dipole moment and $E_e$. When the thickness of PA6 layer is fixed, the $E_0$, $d$ and $E_d$ remain unchanged. Therefore, the primary polarization direction and thickness $D$ ($D_{DFP-o}$ and $D_{DFP-i}$) of ferroelectricity are two crucial factors that significantly affect secondary polarization. The larger the $\theta$, the easier it is for dipoles to go

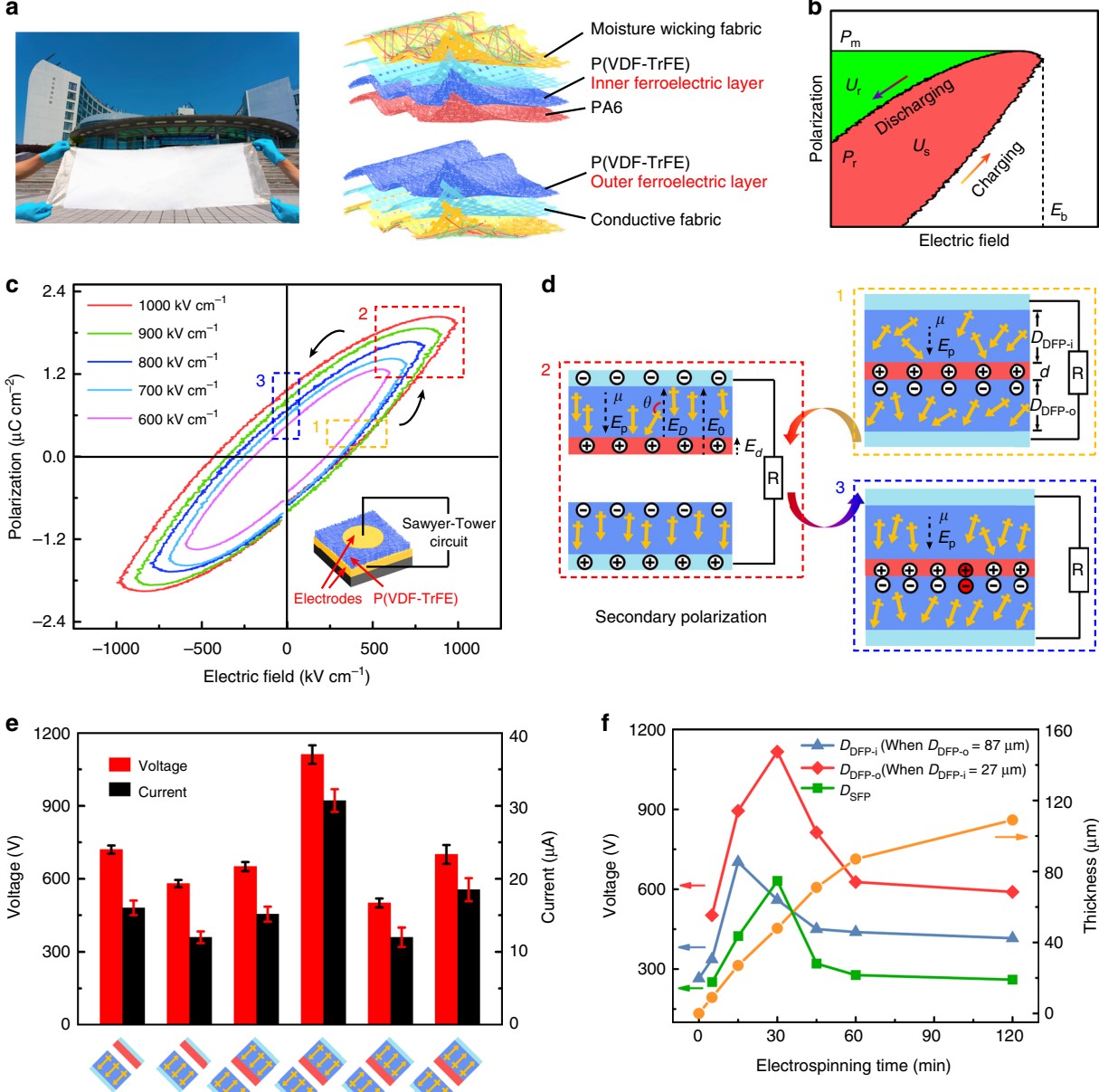

**Fig. 1 The structure of the e-textile and the proposal of tribo-ferroelectric synergy model. a** The physical (one piece of the e-textile 105 × 35 cm) and structural diagram of an all-fiber contact-separation mode tribo-ferroelectric synergistic e-textile. **b** The P-E loop of P(VDF-TrFE) nanofiber ferroelectricity. $U_s$ and $U_r$ are the energy stored in (red area) and released from (green area) ferroelectricity during the change of applied electric field, $P_m$ and $P_r$ are the maximum and remnant polarization intensity, respectively. $E_b$ is the highest electric field a dielectric can sustain. **c** The hysteresis loop of P(VDF-TrFE) nanofiber ferroelectricity under different applied electric fields. **d** Schematic diagram of tribo-ferroelectric synergistic model between ferroelectricity and internal electric field of triboelectric device. $E_O$, $E_D$, $E_d$, $E_p$, $E_e$ are the electric field intensity between dielectric and conductive layer, inside the ferroelectric layer, inside the PA6 layer, generated by the other dipoles and acting on dipoles effectively, respectively. $D$ and $d$ are the thickness of ferroelectric layer and PA6, $\mu$ is dipole moment, $\theta$ is the angle between dipole moment and $E_e$. **e** Influence of the primary polarization direction of P(VDF-TrFE) nanofibers on performance of the e-textile. The error bars correspond to standard deviation caused by the measurement noise. **f** Effect of inner/outer ferroelectric layer thickness on performance of the e-textile. The e-textile tested in experiment were uniformly sized to 4 × 6 cm.

to equilibrium position (state 3), which explains why electrospinning was adopted to prepare P(VDF-TrFE) nanofibers. Similar model for single ferroelectric polarized (SFP) e-textile is illustrated in Supplementary Figs. 4 and 5, and Supplementary Note 1.

The effect of primary polarization direction and thickness on performance of the e-textile was studied (Supplementary Figs. 6 and 7). When the detection between primary polarization and effective electric field $E_e$ change from same to opposite, the output voltage and current rise from lowest to highest (Fig. 1e). Because

the smaller the $\theta$, the more difficult it is for dipoles to reach the equilibrium position (state 3), resulting in a smaller residual polarization and lower performance. The optimal thickness of P(VDF-TrFE) ferroelectricity was investigated by adjusting the electrospinning time. The relationship between its output performance and thickness is shown in Fig. 1f. It can be explained by the fact that the increasing of thickness leads to an opposite trend between the amounts of dipoles and internal electric field intensity.

**Working mechanism of the tribo-ferroelectric synergistic electronics.** To better demonstrate the tribo-ferroelectric synergistic effect, we studied the performance of e-textiles with or without ferroelectricity, i.e., DFP, SFP, and unpolarized (UP) e-textiles. An UP e-textile can be obtained through depolarization of ferroelectric P(VDF-TrFE) (Supplementary Fig. 8, and Supplementary Note 2). The charge density (Fig. 2a), short-circuit current (Fig. 2b), surface potential (Fig. 2c) and output voltage (Supplementary Fig. 9, and Supplementary Note 3) of UP, SFP and DFP e-textiles were tested and compared, respectively. The charge accumulation rate and surface charge density of DFP e-textile are significantly higher than those of SFP and UP e-textiles. We propose, for the first time, a reasonable assumption to demonstrate the effect of ferroelectric polarization on the surface charge transfer between opposite tribo-polarity polymer fibers. It can be summarized that the ferroelectric polarization changes the Fermi level at the surface of tribo-polarity polymers, resulting in a change in surface potential difference[11,24–27]. According to the surface states model[28,29] and ANSYS simulation, as shown in Fig. 2d, a weak surface potential difference is formed between PA6 (tribo-positive polymer) with unpolarized P(VDF-TrFE) (ferroelectric tribo-negative polymer). A small amount of electrons will transfer from PA6 to P(VDF-TrFE) until the Fermi levels of contact surface are equal and reach an equilibrium state. Due to the polarization of P(VDF-TrFE), the Fermi level of contact surface is reduced. While the potential difference between PA6 and P(VDF-TrFE) is increased which enhances the amount of electrons to transfer (Fig. 2e). Furthermore, we introduced an inner ferroelectric layer P(VDF-TrFE) inside of PA6 which results in an increase of Fermi level of PA6. In this case, the surface potential difference between two dielectric materials is further increased and a large number of electrons are transferred from PA6 to P(VDF-TrFE) (Fig. 2f).

We further use the surface states model to illustrate the tribo-ferroelectric synergistic effect during materials contact and separation. When P(VDF-TrFE) and PA6 (two opposite tribo-polarity polymers) first contact, the contact electrification is promoted due to the primary polarization effect. A large amount of electrons are transferred and the Fermi levels of P(VDF-TrFE) and PA6 at contact surface is equal (Fig. 2g Contact state)[11,24–27]. The separation of tribo-polarity materials leads to the internal electric field between dielectric and conductive layer, which promotes secondary polarization of ferroelectricity (Fig. 1c, d State 2) hence changes the Fermi levels of PA6 and P(VDF-TrFE) again (Fig. 2g Separate state). When tribo-polarity materials contact again, the residual polarization of ferroelectricity (Fig. 1c, d State 3) will enhance the capability of capturing charges therefore extra charges will transfer after previous ones (Fig. 2g Next contact state). The Fermi level at the friction interface remains equal, thus completing a cycle. These two effects couple and synergism with each other until an equilibrium is built. Detailed description about charge transfer behavior of tribo-ferroelectric synergistic mechanism is shown in Supplementary Fig. 10, and Supplementary Note 4.

**Construction of all-fiber e-textile with high thermal-moisture stability and comfortability.** In addition to intrinsic mechanism in ideal model, complicated and changeable real-world environment factors are of significance in determining electronics performance. Especially, wet environments are detrimental to electronics. The performance of the triboelectric devices may be greatly suppressed by high humidity and liquid contact[16,17,30]. In order to reduce the negative effects of surface humidity and wet of human body on e-textiles, while maintaining the thermal

equilibrium condition between human body and surrounding environment, the e-textile itself should have the ability of "breathing".

In our multilayer all-fiber electronics, we demonstrate a moisture-wicking fabric based on bilayer hydrophilic nanofiber membranes with different pore sizes and a hydrophobic cotton membrane (Fig. 3a and Supplementary Fig. 11). The functions of each layer are as follows: (1) a hydrophilic polyacrylonitrile (PAN) nanofiber outer layer close to skin which was used to carry away the sweat from human body, (2) a hydrophilic PA6 nanofiber intermediate layer which absorbing sweat from the PAN layer and rapidly diffusing, (3) a hydrophobic and breathable cotton fabric as the inner layer which effectively prevent liquid water in PA6 layer from diffusing to fabric electrode. Since all function layers of the device are built by porous fiber network, the air and moisture can effectively diffuse from skin to environment then exchange heat and moisture with environment (Fig. 3b).

When the human body is in a sweating state (Fig. 3c), the sweat contacts with PAN microfibers, enters interfiber capillary channel and is dragged by the Laplace pressure[31–33]. Whereas the pore size of PAN is about 10 times larger than that of PA6 (Fig. 3d, e), they form a hierarchical network structure from micron to nanometer. The Laplace pressure difference formed at the interface is

$$\Delta P = \frac{4\gamma\cos\theta_{PA6}}{D_{nano}} - \frac{4\gamma\cos\theta_{PAN}}{D_{micro}} \qquad (3)$$

where $\theta_{PA6}$ and $\theta_{PAN}$ are the contact angles (CA) of sweat with PA6 and PAN fibers, $D_{nano}$ and $D_{micro}$ is the pore size of PA6 and PAN fiber layer, $\gamma$ is the surface tension of water in air. The sweat located in the pores of PAN microfibers under the action of $\Delta P$ would penetrate into the pores of PA6 nanofibers. Compared with PAN microfibers, PA6 nanofibers have a higher sweat penetration and spreading driving force as well as larger specific surface area, therefore have faster water evaporation rate (Supplementary Fig. 12b). Furthermore, introduction of PA6 fibers layer can enhance the water uptake capacity of moisture wicking fabric, and also give rise to the wettability gradient between PAN and PA6 layer[34,35]. Thus, more sweat enrichment in PA6 layer will be more conducive to evaporation (Supplementary Fig. 12c, d, and Supplementary Note 5). Figure 3f shows the water evaporation rate of cotton, cotton-PAN and cotton-PA6-PAN fabrics, among which cotton-PA6-PAN fabric has the fastest water evaporation rate. This proves that the hierarchical network structure and high specific surface area of nanofiber composite membranes have directional water transport and moisture wicking performance. Figure 3g demonstrates the penetration and spreading behavior of water droplets on the surface of a moisture-wicking fabric (Supplementary Movie 1). It shows that Laplace pressure facilitates the penetration and spreading of water in cotton-PA6-PAN fabrics. The wicking effect is expected to be helpful for creating comfortable microenvironments to skin.

**Evaluating the wearability and electrical output performance of the e-textile.** We performed various wearability tests on individual functional fabrics and the integrated e-textile (Supplementary Fig. 13). Air permeability refers to the performance of gas molecules through the fabric and is the most basic property in fabric permeability[36]. With the sequential superposition of layers, from moisture-wicking layer (104.67 mm s$^{-1}$), fabric electrode-P(VDF-TrFE) nanofibers layer (84.90 mm s$^{-1}$), fabric electrode-P(VDF-TrFE)-PA6 nanofibers layer (61.30 mm s$^{-1}$) to e-textile (34.10 and 15.90 mm s$^{-1}$), the air permeability shows a gradual

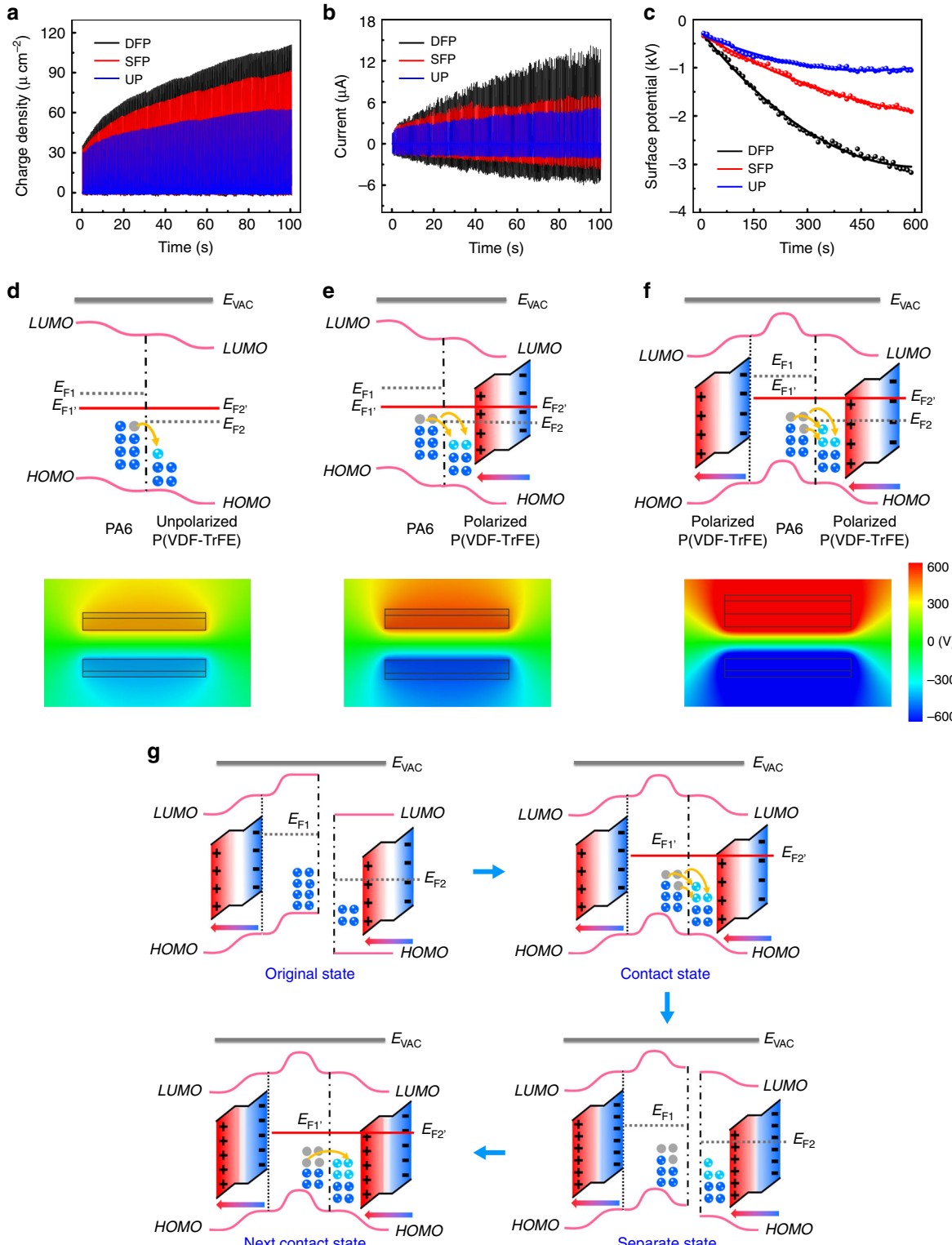

**Fig. 2 Working mechanism of the tribo-ferroelectric synergistic electronics. a** The output charge density and **b** short-current of UP, SFP and DFP e-textiles during contact and separation. **c** The surface potential (tribo-negative materials) versus time of UP, SFP and DFP e-textiles during contact and separation. Surface states model and ANSYS simulation for explaining the effect of ferroelectric polarization on surface charge transfer in **d** PA6 and unpolarized P(VDF-TrFE), **e** PA6 and P(VDF-TrFE) with single ferroelectric polarization effect, **f** PA6 and P(VDF-TrFE) with dual ferroelectric polarization effect during contact and separation. **g** A surface states model for explaining the tribo-ferroelectric synergistic effect between the ferroelectricity and triboelectric internal electric field when e-textile is in contact and separate state. $E_F$, Fermi level; $E_{VAC}$, vacuum level; *LUMO*, the lowest unoccupied molecular orbital; *HOMO*, the highest occupied molecular orbital. $E_{F1}$ and $E_{F2}$ represent the Fermi level (gray dashed line) of PA6 and P(VDF-TrFE) before contact, respectively. $E_{F1'}$ and $E_{F2'}$ represent the Fermi level (red straight line) of PA6 and P(VDF-TrFE) after contact, respectively.

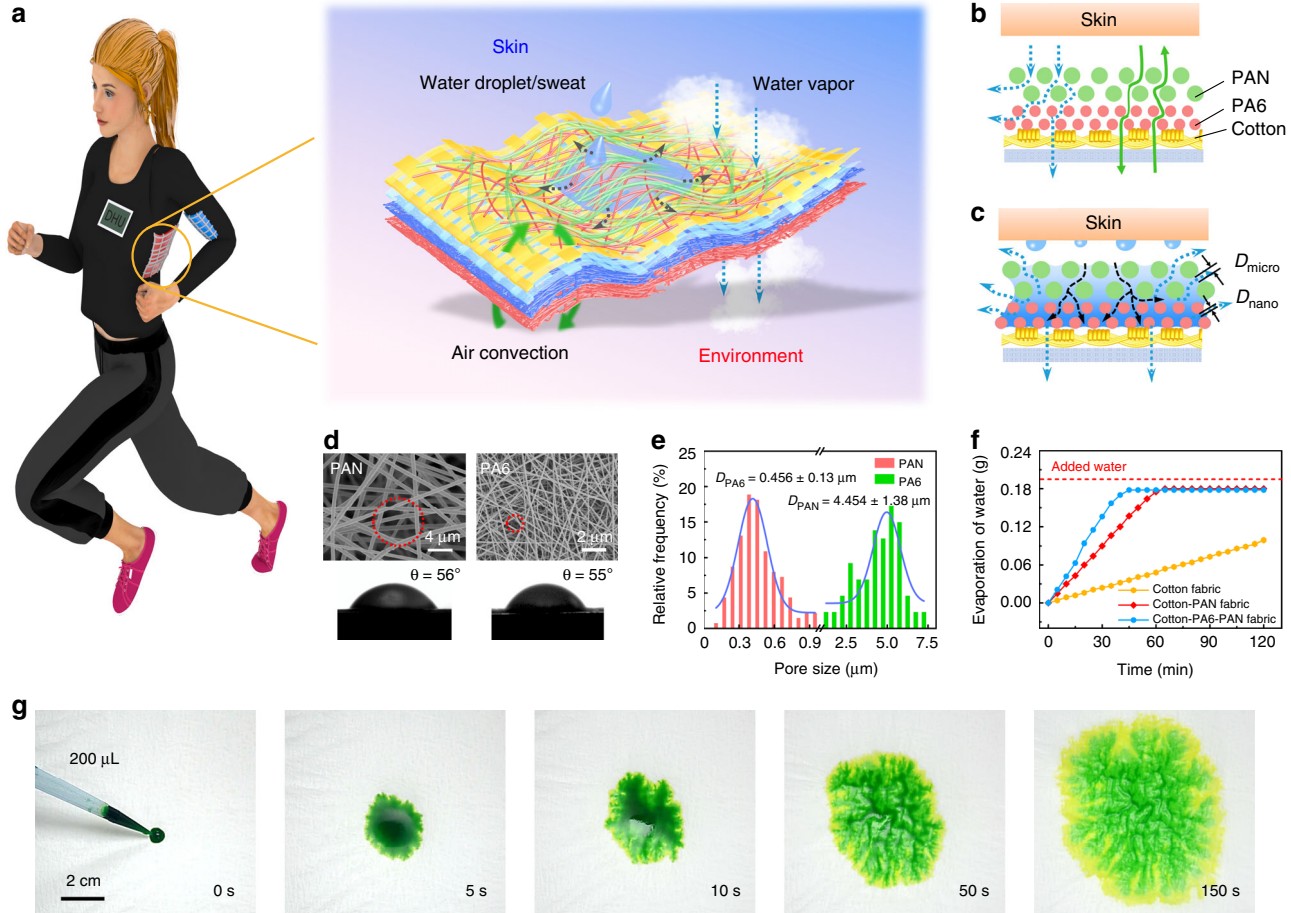

**Fig. 3 Construction of all-fiber e-textile with high thermal-moisture stability and comfortability. a** The e-textile exhibits good wearability which has the functions of breathability, moisture permeability and moisture wicking. **b** Schematic diagram of air permeability, moisture permeability of the e-textile. **c** Schematic diagram of the moisture wicking function of the e-textile in sweating state. From top to bottom, including skin, the moisture-wicking fabric (PAN-PA6-cotton fabric), conductive and dielectric fabric (fabric electrode-P(VDF-TrFE)-PA6). Air convection direction: green arrow. Liquid water transport direction: black dotted arrow. Water vapor transport direction: blue dotted arrow. Wettability gradient: the transition from light blue to dark blue indicates that the water content in the fiber layer changes from less to more. **d** Micrographs, contact angles of hydrophilic PAN and PA6 fibers. **e** Pore size distribution of hydrophilic PAN and PA6 fibers. **f** Water evaporation rate of cotton fabric, cotton-PAN fabric and cotton-PA6-PAN fabric (the moisture-wicking fabric). **g** Wetting behavior (ink droplets, 200 μL) of the moisture-wicking fabric from the top view.

decrease trend (Fig. 4a). But the breathability of integrated e-textile is still higher than commercial jeans (~10 mm s⁻¹)[37]. Because of the presence of numerous macropores in each functional fabric guaranteeing that gas molecules can easily pass through these fiber channels. The moisture permeability of textiles describes the transfer of water/sweat vapor from skin to environment through clothing to maintain the body's heat balance[38]. As shown in Fig. 4b, the moisture permeability of each functional fabric and e-textile were tested (temperature 38 °C, relative humidity 86%). From the moisture-wicking fabric (0.027 g cm⁻²), fabric electrode-P(VDF-TrFE) nanofibers layer (0.023 g cm⁻²), fabric electrode-P(VDF-TrFE)-PA6 nanofibers layer (0.021 g cm⁻²) to e-textile (0.020 and 0.018 g cm⁻²), the obstruction to water/sweat vapor is more obvious.

When human body is in a sweating state, the breathability, moisture permeability and moisture wicking of the garment is important for maintaining the thermal equilibrium condition between human body and surrounding environment[39]. Measurements of thermal and evaporative resistance provided by textile can be used to determine the thermal-moisture comfortability[40]. In thermal resistance test (Fig. 4c, Supplementary Fig. 14, and Supplementary Note 6), to simulate the skin of human body and its surrounding area, the tester consists of three independently

controlled heating zones: test plate, guard ring and lower guard. Each zone is heated to the same temperature (typically 35 °C, close to human skin temperature) to eliminate heat transfer between the different zones. Therefore, all heat loss will only pass through the fabric to surrounding environment (typically 25 °C). In evaporation resistance test, a vapor barrier layer, such as fiberglass paper, was placed between the test plate and the sample to keep the liquid water from wetting the sample.

The average thermal resistance and evaporation resistance of each functional layer are shown in Fig. 4c, where the moisture-wicking fabric (0.040 °C m² W⁻¹, 2.78 Pa m² W⁻¹) and fabric electrode-P(VDF-TrFE) nanofibers layer (0.041 °C m² W⁻¹, 2.9 Pa m² W⁻¹) has low thermal and evaporation resistance. Similarly, as the number of functional fabric layers increases, both thermal and evaporation resistance increase. However, the thermal resistance of two components of e-textile (0.088 and 0.11 °C m² W⁻¹) is still much smaller than 1 clo (unit of thermal resistance defined as the insulation required to keep a resting man comfortable in an environment at 21 °C, air movement 0.1 m s⁻¹ [41]). Above results indicate that the heat generated by human body in normal state can be emitted from skin to outside through e-textile in time without entering sweating state. When human body is sweating, the sweat will be absorbed by the

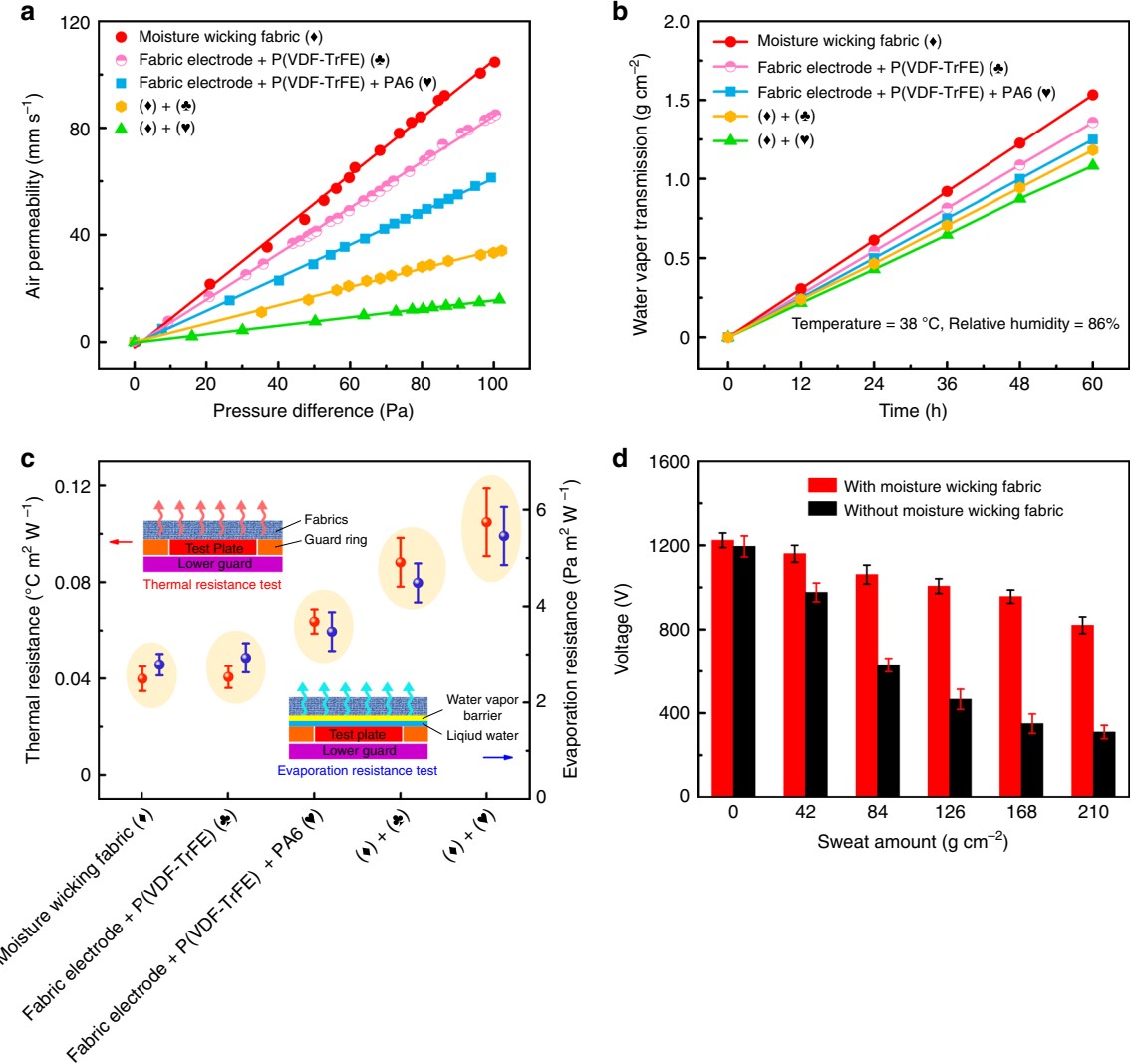

**Fig. 4 Evaluating the wearability and electrical output performance of the e-textile. a** Air permeability test shows the air flow rate through the textile at a pressure difference of 100 Pa on both sides. **b** Water vapor transmission rate test indicates the penetration of sweat or moisture on textiles. **c** Thermal and evaporative resistance test examines the obstacles of heat and moisture flow from skin to environment. The error bars correspond to standard deviation caused by the statistical uncertainty of measurement. **d** The effect of introducing the moisture-wicking fabric on the output performance of e-textile under different sweat amount (simulating human body sweating). When the adult is in exercise (including American football, baseball, basketball, soccer and tennis), the whole-body sweating rates are about 1.21 ± 0.68 L h$^{-1}$[43]. The error bars correspond to standard deviation caused by the measurement noise.

moisture-wicking fabric promptly and releasing to environment in the form of moisture, which makes human body comfortable while reduce the negative effects of wet on e-textiles.

Figure 4d shows the electrical performance of the e-textile under different sweat amount (simulating human body sweating, Supplementary Fig. 15). Due to the barrier effect of hydrophobic cotton fabric (Fig. 3c), a large amount of moisture will preferentially evaporate on both sides and inner surface of the moisture wicking fabric, then only a small portion of water vapor will sequentially penetrate three hydrophobic layers of cotton fabric (CA ≈ 138°), fabric electrode (CA ≈ 113°) and P(VDF-TrFE) (CA ≈ 135°) to the surface of dielectric material. Therefore, the relative humidity on the dielectric material can maintain at a low level (30–50%) and the output voltage (under 100 MΩ) of corresponding e-textile will not be significantly reduced (Supplementary Fig. 16a, b, and Supplementary Note 7). In comparison, without the moisture-wicking fabric, the fabric electrode will directly contact with the skin, causing a large amount of sweat to adhere to the surface of fabric electrode and penetrate into the

friction material. This causes a significant increase in relative humidity (30–83%) and conductivity (due to considerable amount of sodium chloride (NaCl) in sweat)[30] at the friction interface, leading to a significant decrease in contact electrification effect and a large loss of triboelectric charges. In addition, after sweat wicking circles, the output voltage of the e-textile can be basically recovered, but less than the initial output voltage (Supplementary Fig. 16c–e, and Supplementary Note 7). This is due to the influence of residual salt such as NaCl in e-textile. It is noteworthy that after washing and drying, the output voltage of the e-textile can return to the initial value.

**Various applications of the e-textile.** Except for resistance to sweat penetration and moisture, e-textile electrical properties are stable after washing (Supplementary Figs. 17 and 18c, and Supplementary Note 8). The e-textile has good cycle stability and frequency dependence (Supplementary Fig. 18a, b). When the load resistance is 100 MΩ, the e-textile reaches a maximum

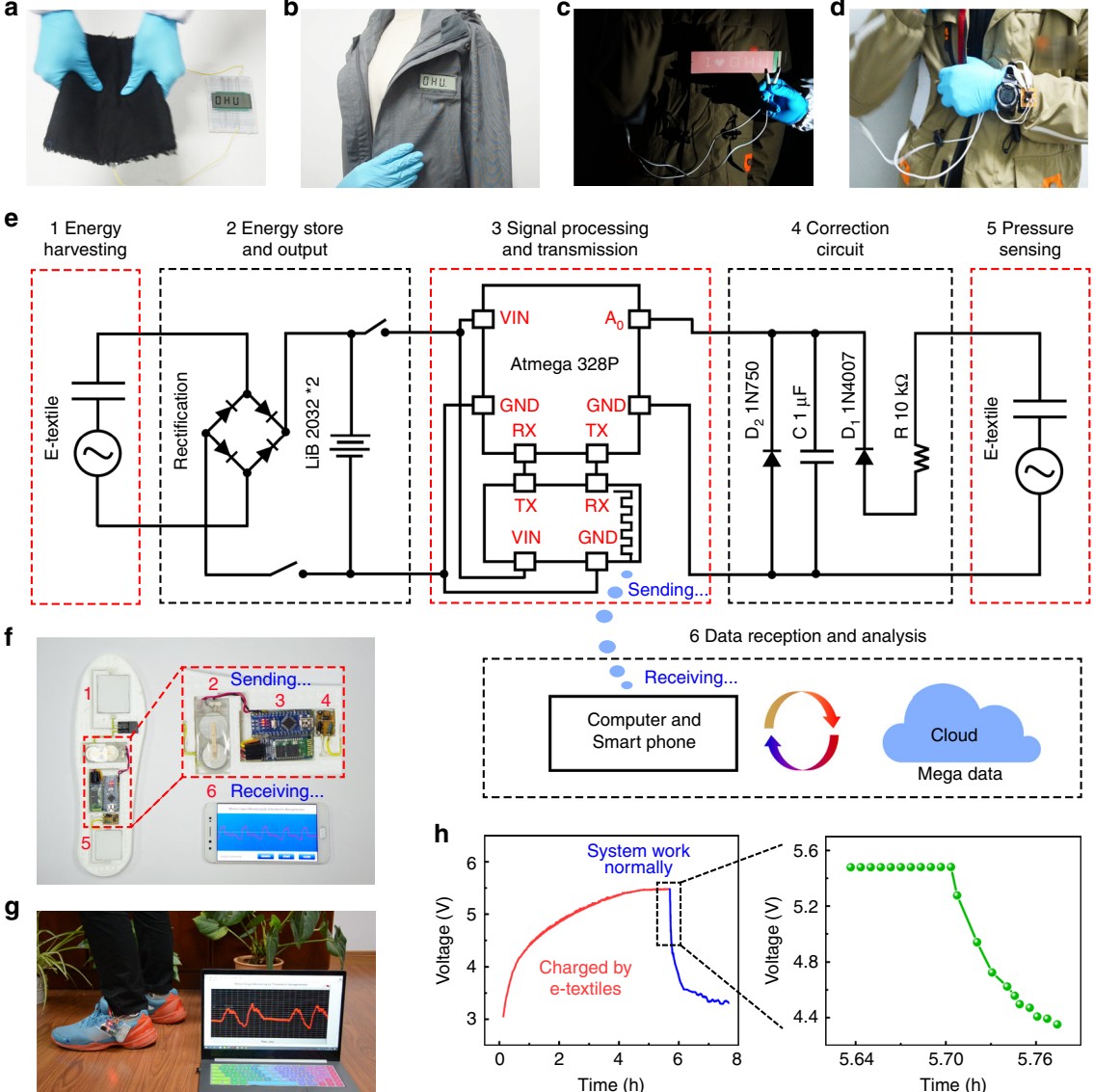

**Fig. 5 Various applications of the e-textile.** The e-textile is **a** sewn together with common fabrics and **b** sewn into clothes to power the LCD by shaking the clothes. E-textile is sewn on the surface of clothes to drive **c** digital electroluminescent lattices and **d** electronic watch by collecting the energy of shoulder movement. **e** E-textiles are used for self-charging, self-sensing gesture monitoring system. The circuit diagram of the self-charging and self-sensing wireless gesture monitoring system. It mainly consists of 6 parts, which are: 1, Energy harvesting; 2, Energy storage and output; 3, Signal processing and transmission; 4, Correction circuit; 5, Pressure sensing; 6, Data reception and analysis. The short distance wireless communication technology was used to send and receive data in real time. **f** Hardware connection of self-powered gesture monitoring system placed on a 3D-printed insole. **g** Self-powered gesture monitoring system captures gait during human movement and transmits it to smartphone or computer in real time. **h** The charge curve of two commercial lithium batteries (LIR 2032) charged by e-textile at a fixed frequency of 2 Hz and the discharge curve of gesture monitoring system during normal operation.

instantaneous power density of $5.20\,\mathrm{W\,m^{-2}}$ which is 7 times more than that of the state-of-the-art breathable moisture-permeable triboelectric textiles (Supplementary Fig. 18d–f, Supplementary Table 1, and Supplementary Note 9). Owing to its all-fiber structure, good flexibility and wearability, the e-textile can be perfectly sewn together with common textiles or directly sewn on existing garments, even placed in a sole (Supplementary Fig. 19a, and Supplementary Movie 2). As shown in Fig. 5a, the e-textile is stitched into common textiles to drive a liquid crystal display (LCD) with "DHU" letters by gently pressing, bending and swaying the textiles. When the e-textile is sewn into clothes, the LCD can be powered by shaking clothes (Fig. 5b). If the e-textile is sewn on the surface of clothes, only the gentle movement of shoulder can drive the pattern of "I ♥ DHU"

consisting of 96 digital electroluminescent lattices which is important for nighttime passive warning (Fig. 5c, and Supplementary Fig. 19b). In addition, when the arm is normally swung, a commercial wearable watch can be powered in 5 s (Fig. 5d, and Supplementary Fig. 19c).

The e-textile also has the characteristic of pressure sensing as evidenced by a free falling impact test[42] (Supplementary Fig. 20, Supplementary Table 2, and Supplementary Note 10). In combination with e-textiles, a self-charging, self-sensing smart insole was developed to monitor the gait of human body in different motion states. The self-powered wireless gesture monitoring system is mainly composed of 6 units: 1, Energy harvesting unit; 2, Energy storage and output unit; 3, Signal processing and transmission unit; 4, Circuit correction unit; 5,

Pressure sensing unit; 6, Data reception and analysis unit (Fig. 5e). E-textiles act as both energy harvesting and pressure sensing unit that functions as a self-powered sensor. The relationship between the units and the process of reading, correcting, wirelessly transmitting, and real-time mapping of pressure sensing signals are explained in detail in Supplementary Fig. 21a, b, and Supplementary Note 11.

The hardware connection and normal operating status of self-powered gesture monitoring system are shown in Fig. 5f, g, and the wireless monitoring system is skillfully placed in a polyurethane insole made by 3D printing. At a fixed frequency of 2 Hz, the e-textile was used to charge two commercial lithium batteries. Within 5.5 h, the battery's voltage increased from 3.01 V to 5.47 V. During the next 1.9 h, the wireless monitoring system continued to work including signal acquisition, processing and transmission. The battery pack quickly discharged and the voltage dropped from 5.47 V to 3.31 V (Fig. 5h). During normal operation of the wireless monitoring system, we performed an electrical signal test on different parts of human foot (Supplementary Fig. 21c, and Supplementary Movie 3). When different parts of the foot are stressed, the contact position and area between tribo-polarity materials will be different, resulting in the respective electrical signals corresponding to the respective parts of the foot. This is expected to be applied to foot motion correction, real-time access to exercise information and prediction of foot ulcers in diabetic patients.

## Discussion

In summary, we set an "all-fiber" principle to design and apply an all-fiber tribo-ferroelectric synergistic electronic with outstanding thermal-moisture stability and comfortability for extracting biomechanical energy. Materials design that used only all-nanofiber materials to constructing tribo-ferroelectric synergistic effect so as to improve output of triboelectric devices. All function layers of the device are built by porous fiber networks to guarantee breathability and moisture permeability. And nanofiber network is constructed into hierarchical structure for realizing moisture wicking function of the e-textile. We discussed the tribo-ferroelectric synergistic mechanism in mechanical-to-electrical energy conversion behavior and moisture wicking mechanism of hierarchical nanofiber network in sweating state. Owing to above designs, the e-textile has a high electrical output as well as outstanding thermal-moisture comfortability. The maximum peak power density of the e-textile can reach 5.2 W m$^{-2}$ under low frequency (~2.5 Hz) motion which is 7 times that of the existing breathable triboelectric textiles. And it's demonstrated to easily power the LCD, digital electroluminescent lattices, electronic watch and monitor motion signals. It is an idea prototypical example for electronic clothing.

## Methods

**Materials**. P(VDF-TrFE) powder (70/30 mol%, $M_w = 1.5 \times 10^4$, Piezotech, France), PA6 pellets ($M_w = 6.0 \times 10^4$, Arkema, France), PAN powder ($M_w = 1.5 \times 10^4$, Formosa Chemicals & Fiber Corporation, China), Hydrophobic breathable cotton fabric (Crystal Shine Red, China), Ni–Cu fabric electrode (X-Silver, China). Dimethylformamide (DMF), Acetone, Formic acid (Aladdin Chemistry Co., Ltd., China).

**Preparation of fabric electrode-P(VDF-TrFE) nanofiber nonwovens**. P(VDF-TrFE) powder was dissolved in a mixture of DMF and acetone (3:2, mass ratio) at 60 °C to prepare the P(VDF-TrFE) solution (20%, mass ratio). P(VDF-TrFE) nanofibers were uniformly deposited on Ni–Cu fabric electrode by electrospinning (TEADFS-700, China). The spinning parameters were as follows: voltage 21 kV, receiving distance 18 cm, propulsion speed 0.8 mL h$^{-1}$, needle inner diameter 0.5 mm.

**Preparation of fabric electrode-P(VDF-TrFE)-PA6 nanofiber nonwovens**. PA6 pellets were dissolved in a mixture of formic acid and acetic acid (4:1, mass ratio) at 60 °C to prepare the PA6 solution (20%, mass ratio). PA6 nanofibers were

uniformly deposited on fabric electrode-P(VDF-TrFE) nonwovens. The spinning parameters were as follows: voltage 20 kV, receiving distance 15 cm, propulsion speed 0.2 mL h$^{-1}$, needle inner diameter 0.2 mm.

**Preparation of the moisture-wicking fabric**. PAN powder was dissolved in DMF at 60 °C to prepare the PAN solution (16%, mass ratio). The PA6 and PAN nanofibers were sequentially deposited onto the hydrophobic breathable cotton fabric by electrospinning. The spinning parameters of PAN were as follows: voltage 15 kV, receiving distance 15 cm, propulsion speed 1.0 mL h$^{-1}$, needle inner diameter 0.5 mm.

**Fabrication of the e-textile**. As shown in Fig. 1a, a fabric electrode-P(VDF-TrFE)-PA6 nanofiber nonwoven fabric was sewn together with the moisture-wicking fabric and wrapped with teflon tape. Similar to the above method, a fabric electrode-P(VDF-TrFE) nanofiber nonwovens and a moisture-wicking fabric were sewn and packaged to obtain another part of the e-textile. Finally, two pieces of fabric were pressed at a pressure of 5 MPa by cold-compacting post treatment to improve the interface bonding between the nanofibers. The e-textile tested in experiment were uniformly sized to 4 × 6 cm (Supplementary Fig. 4).

**Characterization and measurements**. Ferroelectric properties of P(VDF-TrFE) nanofiber nonwovens was tested by Radiant Precision Premier II (Supplementary Fig. 3). Field emission scanning electron microscopy (SEM MERLIN, Carl Zeiss), X-ray diffractometry (D/max-2550VB+, Japan) and Contact Angle Analyzer (OCA40Micro, Germany) were used to characterize the microscopic morphology, crystal phase and the contact angles of nanofibers, respectively. The Keithley 2657 A and Keithley 6514 were used to test the electrical output performance of the e-textile. The surface potentials of the e-textile were determined using an electrostatic voltmeter (TREK 542A-2, USA), at a relative humidity of ~30%. Relative humidity was tested by humidity measuring instrument (GM620, Shanghai Tianzhi Intelligent Technology Co., Ltd., China). The water evaporation rate was tested by moisture evaporation rate tester (FFZ191, Wenzhou Fangyuan Instrument Co., Ltd., China) based on GB/T 21655.1−2008 standard (Supplementary Fig. 13b). The fabric permeability to air was measured by the air permeability tester (YG461E, Wenzhou Fangyuan Instrument Co., Ltd., China) followed by GB/T 24218.15-2018 standard (Supplementary Fig. 13a). Water vapor transmission rate test was measure by fabric moisture permeability testing apparatus (YG601H, Ningbo Textile Instrument Factory, China) followed by GB/T 12704.1-2009 standard (Supplementary Fig. 13c). The thermal and evaporation resistance was assessed according to ASTM F1868 standard using a sweating guarded hotplate (SGHP, Northwest Testing Technology Corporation, US) (Supplementary Fig. 14, and Supplementary Note 6).

## Data availability

The data that support the findings of this study are available from the corresponding authors upon reasonable request.

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

## Acknowledgements

We gratefully acknowledge the financial support by the Fundamental Research Funds for the Central Universities (2232019A3-02), DHU Distinguished Young Professor Program (LZB2019002), Natural Science Foundation of China (No. 51603037), and the Innovation Program of Shanghai Municipal Education Commission (2017-01-07-00-03-E00055). C.H. thanks the Young Elite Scientists Sponsorship Program by CAST (2017QNRC001). W.G. thanks the Graduate Student Innovation Fund of Donghua University (CUSF-DH-D-2019008).

## Author contributions

C.H., Q.Z., and H.W. guided the project. W.Y., W.G., C.H., and H.W. conceived the idea and designed the experiment. W.Y., Y.S., Y.G., and W.Z. performed the experiments and measurements. Y.L. revised the manuscript. All authors analyzed the experimental data, drew the figures and prepared the manuscript. All authors discussed the results and reviewed the manuscript.

## Competing interests

The authors declare no competing interests.
