## [Peer Review File · Nature Communications]

Reviewers' comments:

Reviewer #1 (Remarks to the Author):

In this paper, authors reported a new comprehensive electronic textiles which can transfer mechanical energy to electric energy through the tribo induced charges from two ferroelectric layers with several polarization. The electronic textiles were covered by moisture wicking fabric which enable the whole electronic textiles have nice performance even under high moisture atmosphere. Besides, the electronic textiles have good thermal-moisture comfortability. This work is very interesting and important to the field. Before the manuscript can be accepted by Nature Communications, there are some issues should be addressed first:

1. Does the authors measured the electronic textiles without poled the ferroelectric layer? Or only poled one ferroelectric layer? Authors should compared these results to further prove their mechanism is correct.
2. All Equations in manuscript and supporting information are not clear.
3. Authors declared the performance of the electronic textiles has 7 times larger than previous reported ones, whether the measurement conditions are the same?
4. Authors should show the current of the triboelectric signal for their electronic textiles as well.

Reviewer #2 (Remarks to the Author):

The author realized all-fiber tribo-ferroelectric synergistic e-texture with thermal-moisture stability and comfortability. They discussed the tribo-ferroelectric synergistic mechanism in mechanical-to-electrical energy conversion behavior and moisture wicking mechanism of hierarchical nanofiber network in sweating state. Using ferroelectric polymer nanofibers showed peak power density of 5.2 W/m², and breathability increased by 7 times. The manuscript was written systematically and logically. However, it needs to improve some of points. I would like to recommend the manuscript to be published at Nature Communication after the revision.

- 1) In Figure 1e and 1f, authors should confirm the performance of the device without inner ferroelectric layer experimentally, even though they analyzed this by simulation.
- 2) They need to mention the device size what they used in Figure 1e. 4×6 cm.?
- 3) In this paper, tribo-ferroelectric synergistic mechanism is one of the key factors to explain the high performance of this hybrid system. They need to explain the charge behavior when it contacts each other. For example, the reasons why the amounts of the charge transfer are different between first contact and second contact. They have drawn the schematic without any evidence experimental and simulation.
- 3) In Figure 3b and 3c, the PA6 and PAN have similar surface polarities measured by contact angle. And, PAN has 10 times larger pore size. It is necessary to explain why the PA6-PAN can evaporate the wafer more effectively, even though the PA6-PAN fabric has much thicker and much smaller pore size than the PAN fabric.
- 3) In Figure 4d, the device performance should be supported by cyclic data over time. How many times is the data? Over time, the system should monitor the durability of the device by moisture whether or not it has moisture wicking fabric. For example, even if there is no moisture wicking fabric in the device, it is necessary to compare whether the device performance can be restored to its original state or continuously deteriorated as the water evaporates.

Reviewer #3 (Remarks to the Author):

The manuscript " All-fiber tribo-ferroelectric synergistic electronics with high 2 thermal-moisture stability and comfortability " suggested the use of tribo-ferroelectric fiber to improve triboelectric output performance and good thermal-moisture stability of electronic clothing. Using ferroelectric

polymer nanofibers showed peak power density of 5.2 W/m², and breathability increased by 7 times. Besides, material properties and specific functions of each layer are well described, and various examples for applications using their electronic clothing are suggested. An interesting manuscript could be published. The reviewer has a few minor comments as below.

1. Need to describe the correct mechanism for triboelectric effect of tribo-ferroelectric synergistic electronics in Fig. 2. As considering working principle of TENG, the output performance is determined by the extent of the exchange of charges. However, there is no experimental result to prove working mechanism that triboelectric performance change step by step.
2. Need more detailed descriptions about moisture-wicking effect for electronic fabric. How can sweat be transpired out of a fabric without significantly reducing the triboelectric output voltage?

Responses to the Reviewers' Comments

Point-by-point responses to the reviewers' comments

We sincerely thank the reviewers for their careful and thorough review, which are indeed very helpful to make the paper more solid and smooth. We have revised our manuscript very carefully in the light of their suggestions and comments.

The following responses have been prepared to address all of the reviewers' comments in a point-by-point fashion. (Comments in black, responses in blue):

Response to Reviewer #1

General comment: *In this paper, authors reported a new comprehensive electronic textile which can transfer mechanical energy to electric energy through the tribo-induced charges from two ferroelectric layers with reversal polarization. The electronic textiles were covered by moisture wicking fabric which enable the whole electronic textiles have nice performance even under high moisture atmosphere. Besides, the electronic textiles have good thermal-moisture comfortability. This work is very interesting and important to the field. Before the manuscript can be accepted by Nature Communications, there are some issues should be addressed first.*

Response: Thanks for your review and we appreciate your feedback. We have carefully revised the manuscript according to your comments. The replies to each of your concern are listed below.

1. Does the authors measured the electronic textiles without poled the ferroelectric layer? Or only poled one ferroelectric layer? Authors should compare these results to further prove their mechanism is correct.

Response: Thank you very much for your professional advice. We have conducted additional experiments on the electrical output of single-polarized ferroelectric (SFP) e-textile. In addition, based on the "depolarization" idea of P(VDF-TrFE) ferroelectricity by heat treatment, an unpolarized ferroelectric (UP) e-textile was also designed and measured. The charge density, short-circuit current, surface potential and voltage (under 100 M Ω load) of UP, SFP and dual ferroelectric polarized (DFP) e-textiles were compared, the corresponding results further verify the tribo-ferroelectric synergistic mechanism.

● The electrical output performance of SFP e-textile.

The effect of primary polarization direction (θ) and thickness (D_{SFP}) of ferroelectricity on the

1 performance of SFP e-textile was studied. When the detection between primary polarization and
 2 effective electric field ($E_D + E_p$) change from same to opposite, the output voltage and current are
 3 significantly improved (Figure R1b). Because the smaller the θ , the more difficult it is for dipoles to
 4 reach the equilibrium position, resulting in a smaller residual polarization and lower performance.
 The relationship between output performance and thickness is shown in Figure R1c, and d. The results
 can be explained by the fact that the increasing of thickness leads to an opposite trend between the
 amounts of dipoles and internal electric field intensity.

 **Figure R1.** (a) Schematic diagram of tribo-ferroelectric synergistic model in single ferroelectric polarization (SFP)
 e-textile. (b) Influence of primary polarization direction (θ) of P(VDF-TrFE) nanofibers on the performance of SFP
 e-textile. (c), and (d) Effect of ferroelectric layer thickness (D_{SFP}) on voltage (under 100 M Ω load) and short-circuit
 current of SFP e-textile.

 ● **Preparation of unpolarized ferroelectric P(VDF-TrFE) nanofiber nonwovens**

It is well known that the electrospinning process has in situ poling effect and therefore induces
 preferred dipole orientation in P(VDF-TrFE) nanofiber ferroelectricity [1]. In order to prepare an
 unpolarized P(VDF-TrFE) nanofiber nonwovens that has the same microstructure as the originally
 polarized nanofiber ferroelectricity, we proposed the idea of “depolarization” (Figure R2a). The
 ideas and experimental results are discussed as follows.

**Figure R2. Depolarization of electrospun P(VDF-TrFE) nanofiber nonwovens.** (a) Schematic diagram of
 depolarization of P(VDF-TrFE) ferroelectricity by heat treatment. (b) Micrographs of P(VDF-TrFE) nanofibers
 after different heat treatment temperatures. (c) Measurement of piezoelectric coefficient (d_{33}) of depolarized P(VDF-
 TrFE) nanofiber nonwovens. (d) As the heat treatment temperature increases, the piezoelectric coefficient (d_{33}) of
 P(VDF-TrFE) ferroelectricity is continuously lowered to achieve the depolarization effect. (e) Circuit diagram for
 piezoelectric coefficient (d_{33}) test.

For piezoelectric and ferroelectric materials, the depolarization can be achieved by heat treatment,
 applying reverse voltage and so on [2]. The degree of polarization and depolarization can be measured
 by the piezoelectric coefficient d_{33} (d_{33} : Proportional constant of the linear response relationship

between mechanical and electrical quantities [1]). Here, we treated the electrospun P(VDF-TrFE)
nanofiber nonwovens at different temperatures to study the depolarization behavior and
microstructure changes (Figure R2b, c, and d). As the heat treatment temperature increases, the d_{33} of
P(VDF-TrFE) ferroelectricity decreases continuously. When temperature reaches 190 °C, the
nanofibers begin to melt and the microscopic morphology changes, which would change the surface
friction behavior and affect the charge transfer process. When the temperature is higher than 180 °C,
the d_{33} value does not decrease significantly, which indicates that the heat treatment at 180 °C for 3 h
is a suitable depolarization process of P(VDF-TrFE) without significantly changing its microstructure.
Figure R2e shows the circuit diagram of the d_{33} test. Therefore, the P(VDF-TrFE) obtained by heat
treatment at 180 C for 3 hours can be considered to have completely depolarized. This sample was
use in the preparation of unpolarized ferroelectric (UP) e-textile for control experiments.

● Comparison of the electrical output performance of UP, SFP and DFP e-textiles.

To further demonstrate the enhancement of ferroelectric effect on triboelectric textiles, the output
charge density (Figure R3a), short-circuit current (Figure R3b, and f), surface potential (Figure R3c,
15 d, and e) and voltage (under 100 M Ω) (Figure R3f) of UP, SFP and DFP e-textiles were tested and
16 compared, respectively. Figure R3a shows the process of charge accumulation in three e-textiles step
by step. The UP e-textile reaches a maximum value of $\sim 106 \mu\text{C}\cdot\text{m}^{-2}$ only after 1100 cycles operation,
while the SFP and DFP e-textiles increase with a much bigger slope and reach the maximum value of
$\sim 161 \mu\text{C}\cdot\text{m}^{-2}$ after 2800 cycles and $\sim 320 \mu\text{C}\cdot\text{m}^{-2}$ after 6600 cycles. Similarly, with the continue of
contact-separate processes, the DFP e-textile has the fastest growth rate and the highest saturation
current ($\sim 33 \mu\text{A}$). The current growth rate and saturation current ($\sim 16 \mu\text{A}$) of SFP e-textile are higher
than those of UP e-textile ($\sim 10 \mu\text{A}$).

In addition, during the contact-separation process, we also tested the surface potential of tribo-
negative material (P(VDF-TrFE)). The surface potential was detected every 10 seconds, and the
distance between probe and the textile was fixed at 1.5 cm (Figure R3c). In 600 seconds, the surface
potential of P(VDF-TrFE) in UP e-textile increased from -0.3 kV to -1.05 kV, while the SFP and DFP
textiles increased from -0.34 kV and -0.29 kV to -1.91 kV and -3.1 kV, respectively. It visually shows
the accumulation of triboelectric charges step by step. And the charge accumulation rate and charge
density at the friction interface were significantly improved due to the tribo-ferroelectric synergistic
enhancement mechanism. As shown in Figure R3f, the output voltage results have similar trend.

In summary, we have compared the surface potential (surface charges), charge density (induction
charges), short-circuit current and voltage of UP, SFP and DFP e-textiles, the corresponding results
strongly prove the proposed effect of ferroelectric polarization on surface charges transfer between
opposite tribo-polarity polymer fibers.

 **Figure R3.** (a) The output charges density of UP, SFP and DFP e-textiles working continuously for 7200 cycles at
 a fixed frequency of 2.5 Hz. (b) The short-current of UP, SFP and DFP e-textiles working continuously for 7200
 cycles at a fixed frequency of 2.5 Hz. (c) Digital photo of the surface potential test device for e-textiles. (d)
 Schematic diagram of surface potential test procedure for UP, SFP and DFP e-textiles. (e) The surface potential
 (tribo-negative materials) versus time of UP, SFP and DFP e-textiles during contact and separation. (f) Comparison
 of output voltage (under 100 M Ω load) and short-circuit current of UP, SFP and DFP e-textiles at a fixed frequency
 of 2.5 Hz.

**● Our revision to the manuscript:**

(i) We added “in dual-ferroelectric polarized (DFP) e-textile” (Please see page 5) and “Similar model
 for single ferroelectric polarized (SFP) e-textile is illustrated in Supplementary Fig. 5.” (Please see
 page 6) in the revised manuscript.

We added in the legend to Figure1d “ E_{F1} and E_{F2} represent the Fermi level (gray dashed line) of
 PA6 and P(VDF-TrFE) before contact, respectively. E_{F1}' and E_{F2}' represent the Fermi level (red

straight line) of PA6 and P(VDF-TrFE) after contact, respectively.” (Please see page 8) in the revised
manuscript.

We added “To better demonstrate the tribo-ferroelectric synergistic effect, we studied the
performance of e-textiles with or without ferroelectricity, i.e., DFP, SFP, and unpolarized (UP) e-
textiles. An UP e-textile can be obtained through depolarization of ferroelectric P(VDF-TrFE)
(Supplementary Fig. 8, and Supplementary Note 2). The charge density (Fig. 2a), short-circuit current
(Fig. 2b), surface potential (Fig. 2c) and output voltage (Supplementary Fig. 9 and Supplementary
Note 3) of UP, SFP and DFP e-textiles were tested and compared, respectively. The charge
accumulation rate and surface charge density of DFP e-textile are significantly higher than those of
SFP and UP e-textiles.” (Please see page 8) in the revised manuscript.

We also added “The Keithley 2657A and Keithley 6514 were used to test the electrical output
performance of the e-textile. The surface potentials of the e-textiles were determined using an
electrostatic voltmeter (TREK 542A-2, USA), at a relative humidity of approximately 30 %
(Supplementary Fig. 9c).” (Please see page 20) in experimental section of the revised manuscript.

Corresponding changes have been marked in red in the revised manuscript.

(ii) We added the polarization direction and thickness data (Red wireframe in Figure R4b, c) of the
ferroelectric layer in SFP e-textile in Figure 1e, and f. We adjusted the subscript ($D_o \rightarrow D_{DFP-o}$ and D_i
$\rightarrow D_{DFP-i}$) of the thickness of ferroelectric layer (Red wireframe in Figure R4 a) in Figure 1d. (Please
see page 4)

**Figure R4.** Mark of the Corresponding changes in original manuscript Figure 1. (a) Figure R4a corresponds
 to Figure 1d in the original manuscript. (b) Figure R4b corresponds to Figure 1e in the original manuscript. (c)
 Figure R4c corresponds to Figure 1f in the revised manuscript.

**(iii)** We also added the comparison data of charge density, short-circuit current and surface potential
 of UP, SFP and DFP e-textiles during contact separation as Figure 2 a, b, and c. (Please see page 7)

**Figure R5.** Comparison data of charge density, short-circuit current and surface potential during contact-separation
 process of UP, SFP and DFP e-textiles were added as Figure 2 a, b, and c.

**• Our revision to the supplementary materials:**

**(i)** We added discussions of tribo-ferroelectric synergistic enhancement model in single ferroelectric
 polarized (SFP) e-textile as Note S1: “When the detection between primary polarization and effective
 electric field ($E_D + E_p$) change from same to opposite, the output voltage and current are significantly

improved (Figure S5b). Because the smaller the θ , the more difficult it is for dipoles to reach the
equilibrium position, resulting in a smaller residual polarization and lower performance. The
relationship between output performance and thickness is shown in Figure S5c, and d. The results can
be explained by the fact that the increasing of thickness leads to an opposite trend between the
amounts of dipoles and internal electric field intensity.”.

We added the design idea and preparation method of “depolarization” for electrospun nanofiber
nonwovens as Note S2: “In order to prepare an unpolarized P(VDF-TrFE) nanofiber nonwovens that
has the same microstructure as the originally polarized nanofiber ferroelectricity, we proposed the
idea of “**depolarization**” (Figure S8a). Here, we treated the electrospun P(VDF-TrFE) nanofiber
nonwovens at different temperatures to study the depolarization behavior and microstructure changes
(Figure S8b, c, and d). As the heat treatment temperature increases, the d_{33} of P(VDF-TrFE)
ferroelectricity decreases continuously. When temperature reaches 190 °C, the nanofibers begin to
melt and the microscopic morphology changes, which would change the surface friction behavior and
affect the charge transfer process. When the temperature is higher than 180 °C, the d_{33} value does not
decrease significantly, which indicates that the heat treatment at 180 °C for 3 h is a suitable
depolarization process of P(VDF-TrFE) without significantly changing its microstructure. Figure S8e
shows the circuit diagram of the d_{33} test. Therefore, the P(VDF-TrFE) obtained by heat treatment at
180 °C for 3 hours can be considered to have completely depolarized. This sample was use in the
preparation of unpolarized ferroelectric (UP) e-textile for control experiments.”.

We also added the discussion of electrical output performance of UP, SFP and DFP e-textiles as
Note S3: “Figure S9a shows the process of charge accumulation in three e-textiles step by step. The
UP e-textile reaches a maximum value of $\sim 106 \mu\text{C}\cdot\text{m}^{-2}$ only after 1100 cycles operation, while the
SFP and DFP e-textiles increase with a much bigger slope and reach the maximum value of ~ 161
$\mu\text{C}\cdot\text{m}^{-2}$ after 2800 cycles and $\sim 320 \mu\text{C}\cdot\text{m}^{-2}$ after 6600 cycles. Similarly, with the continue of contact-
separate processes, the DFP e-textile has the fastest growth rate and the highest saturation current
($\sim 33 \mu\text{A}$). The current growth rate and saturation current ($\sim 16 \mu\text{A}$) of SFP e-textile are higher than
those of UP e-textile ($\sim 10 \mu\text{A}$). In addition, during the contact-separation process, we also tested the
surface potential of tribo-negative material (P(VDF-TrFE)). It was detected every 10 seconds, and
the distance between probe and the textile was fixed at 1.5 cm (Figure S9c). In 600 seconds, the
surface potential of P(VDF-TrFE) in UP e-textile increased from -0.3 kV to -1.05 kV, while the SFP
and DFP textiles increased from -0.34 kV and -0.29 kV to -1.91 kV and -3.1 kV, respectively. It
visually shows the accumulation of triboelectric charges step by step. And the surface charge
accumulation rate and charge density at the friction interface were significantly improved due to the
tribo-ferroelectric synergistic enhancement mechanism. As shown in Figure S9d, the output voltage
results have similar trend.”.

(ii) Figure R1 and R2 were added as Figure S7 and S8, respectively. Figure R3a (i, iii), b (i, iii), c (i,
ii), d and f were added as Figure S9.

2. All Equations in manuscript and supporting information are not clear.

**Response:** Thank you very much for your careful review. We have re-formatted the equations in
manuscript and supporting information to ensure they can be clearly read. In addition, the equations
have been deduced in detail to ensure that they can be well understood.

● **Derivation of Equation (1):**

According to thermodynamic relationships for dielectrics (Figure R6a) [3], when a cyclic electric
field is applied on the dielectric and induces an electric displacement polarization, the energy density
store in P(VDF-TrFE) ferroelectricity is

$$U_s = U - U_r = \int_0^{P_m} E dP - \int_{P_r}^{P_m} E dP \quad (1)$$

where U is the energy for charging ferroelectricity during the increase of electric field, U_s and U_r are
the energy stored in and released from ferroelectricity during the change of applied electric field. P
and E are the electric displacement polarization and electric field intensity, respectively. P_m and P_r
are the maximum and remnant polarization intensity, respectively. The breakdown strength (E_b)
indicates the highest electric field for the dielectrics which is the critical parameter to defines the
maximal energy density achievable [4].

● **Derivation of Equation (2):**

The effective electric field intensity E_e received by the CF_2 dipole (Figure R6b) includes the electric
field intensity E_D formed by surface polarization charge and the electric field intensity E_p formed by
the other dipoles inside ferroelectric layer, which can be expressed as

$$E_e = E_D + E_p = \left(E_D - \frac{d}{D} \cdot E_d \right) + E_p \quad (2)$$

where D and d are the thickness of ferroelectric layer and PA6 layer.

The energy required for dipoles rotating to equilibrium position [5], [6] can be expressed by the
following formula:

$$\Delta U = \int_{\theta_1}^{\theta_2} \mu E \sin \theta d\theta = \mu E_e (\cos \theta - (\cos 180^\circ)) = \mu E_e (\cos \theta + 1) \quad (3)$$

where μ is dipole moment, θ is the angle between CF_2 dipole moment and E_e .

Combining formulas (2) and (3), the energy required for CF_2 dipole steering can be expressed as

$$\Delta U = \mu \left(E_D - \frac{d}{D} \cdot E_d + E_p \right) (\cos \theta + 1) \quad (4)$$

According to Equation (4), when the thickness of the PA6 film is fixed, the primary polarization
direction (θ) of CF_2 dipole and thickness (d) of the P(VDF-TrFE) ferroelectricity are two crucial
factors that affect the secondary polarization.

● **Derivation of Equation (3):**

When moisture wicking fabric is in contact with sweat (Figure R6c), the sweat infiltrates the
hydrophilic nanofiber fabric, and capillary force (Laplace pressure) of fiber channel can be expressed
as [5-7]

$$9 \quad P = \frac{4\gamma\cos\theta}{D} \quad (5)$$

where θ is the contact angle between water and the fabric, D is the pore diameter composed of
nanofibers, γ is the surface tension of water in air. According to Equation (5), the Laplace pressure is
inversely proportional to D . For PAN and PA6 fabrics, the capillary force of the channel can be
express as

$$14 \quad P_{PAN} = \frac{4\gamma\cos\theta_{PAN}}{D_{PAN}},$$
$$15 \quad P_{PA6} = \frac{4\gamma\cos\theta_{PA6}}{D_{PA6}} \quad (6)$$

16 When sweat flows through the interface between PAN and PA6 fibers, the pressure difference ΔP
17 between the two is

$$18 \quad \Delta P = P_{PA6} - P_{PAN} = \frac{4\gamma\cos\theta_{PA6}}{D_{PA6}} - \frac{4\gamma\cos\theta_{PAN}}{D_{PAN}} \quad (7)$$

**Figure R6.** (a) The P-E loop of P(VDF-TrFE) nanofiber ferroelectricity, demonstrating stored energy density U_s ,
 released (discharged) energy density U_r and charged energy density U ($U=U_s+U_r$). (b) Schematic diagram of tribo-
 ferroelectric synergistic model between ferroelectricity and internal electric field of triboelectric device. (c)
 Schematic illustration of the directional water transport mechanism and Laplace pressure in moisture wicking fabric.

**● Our revision to the manuscript:**

We have carefully revised all equations and made necessary corrections. The equation 1 has been
 modified to:

$$U_s = U - U_r = \int_0^{P_m} E dP - \int_{P_r}^{P_m} E dP \quad (1)$$

And we also added “ U is the energy for charging ferroelectricity during the increase of electric field,”
 in the revised manuscript. (Please see page 5)

We modified the format of all the equations in the revised manuscript (Formula 2: Please see page
 6, Formula 3: Please see page 11):

$$\Delta U = \mu \left(E_D - \frac{d}{D} \cdot E_d + E_p \right) (\cos \theta + 1) \quad (2)$$

$$\Delta P = \frac{4\gamma \cos \theta_{PA6}}{D_{nano}} - \frac{4\gamma \cos \theta_{PAN}}{D_{micro}} \quad (3)$$

Corresponding changes have been marked in red in the revised manuscript.

● **Our revision to the supplementary materials:**

We modified the format of all the equations in supporting information (Formula 1, 2, 3 and 4: *Please*
*see page 18*, Formula 5 and 6: *Please see page 19*):

$$R_{ct} = \frac{T_{skin} - T_{amb}}{Q/A} \quad (1)$$

$$R_{et} = \frac{P_{skin} - P_{amb}}{Q/A} \quad (2)$$

$$P_{amb} = RH \cdot 133.3 \cdot 10 \exp[8.10765 - (1750.29/(235 + T_{amb}))] \quad (3)$$

$$P_{skin} = 133.3 \cdot 10 \exp[8.10765 - (1750.29/(235 + T_{skin}))] \quad (4)$$

$$\frac{1}{2}mv^2 = mgh \quad (5)$$

$$m(v_{n+1} + v_n) = F_n T_A \quad (6)$$

3. *Authors declared the performance of the electronic textiles has 7 times larger than previous*
*reported ones, whether the measurement conditions are the same?*

**Response:** Thank you for your professional comments. In order to make our statement more rigorous
and persuasive, we systematically classify and organize measure conditions and electrical parameters
for work in the field of triboelectric textiles. s to the best our knowledge. These energy textiles are
mainly divided into three categories: All-yarn (breathable), All-fabric (breathable) and Airtight
(impermeable) as shown in Table 1. It should be noted that the voltage and current listed in table are
the corresponding outputs of peak power density. Several typical comparisons are listed below.

Reference [Guo, Yinben, et al. Nano Energy 48 (2018): 152-160.] is our previous work on all-
fabric breathable triboelectric textiles with a peak power density of 0.697 W/m². To the best of our
knowledge, the power density reported in this reference was the highest value in the field other than
this work, while the output power density of our tribo-ferroelectric synergistic electrics reported in
this work is 7 times higher than it. The measurement conditions, such as frequency and humidity, are
similar to this reference. The test results of our e-textile at 2 Hz are shown in Figure S18a.

Reference [Gong, Wei, et al. Nature communications 10.1 (2019): 868.] is our previous work on
all-yarn breathable triboelectric textiles. Its operating frequency of 0.7 Hz is one quarter of ours, but
our output power density is about 741 times higher.

Reference [Pu, Xiong, et al. Advanced Materials 27.15 (2015): 2472-2478.] reports a triboelectric
fabric based on Ni-cloth and parylene-cloth, which has an output power density of 0.393 W/m² at an
operating frequency of 0.7 Hz. Although its operating frequency is one-third of ours, our output power
density is 12.2 times higher than it.

In addition, in the category of airtight triboelectric textiles, the output power density of our tribo-
ferroelectric synergistic electronics is still at a higher output level.

1

2 **Table R1 Summary of measurement conditions and electrical parameters of published triboelectric textiles.**

	Mode	Tribo-materials	Frequency /Hz	Relative humidity /%	Contact area /cm ²	Voltage / V	Current / μ A	Power density /W·m ⁻²	Refs.
All-yarn	CS	Silicone rubber & Stainless steel	0.5	/	/	60	/	~0.007	[1]
	SE	Polyester & Stainless-steel	/	/	36	~75	~1.2	~0.06	[2]
	SE	Polyester & Stainless steel	3	/	16	~100	~0.72	~0.085	[3]
	CS	PDMS & Stainless steel	3	/	2.25	~35	~0.79	0.263	[4]
All-fabric	SE	PET-PVDF-PTFE & Common fabric	3	/	16	112.7	~1.7	0.08	[5]
	SE	PET & Water	/	/	2.25	8	~2.6	0.14	[6]
	FT	PTFE & Common fabric	/	~15	45	~200	~4	0.203	[7]
	CS	Parylene & Ni	0.7	/	25	50	~4.25	0.393	[8]
	CS	PVDF & Silk	2	~30	8	78	~8.2	0.697	[9]
	CS	P(VDF-TrFE) (With two ferroelectric layers) & PA6	2	~30	24	1116	~13.7	5.2	Our work
Airtight, impermeable	CS	Polyethersulfone + carbon black + polystyrene & Cellulose acetate	3	/	9	115.2	~1.5	0.13	[10]
	CS	PVDF & Acetate + polyurethane	3	~55	/	345	9.8	1.3	[11]
	CS	PTFE + PVDF + EVOH & Copper	10	/	16	170.5	~14	2.45	[12]
	CS	PVDF & PHBV	2	~20	22	695	~10.1	3.1	[13]
	SE	Hydrophobic cellulose oleoyl ester nanoparticles + black phosphorus & Skin	4	/	49	880	14.7	5.1	[14]
	CS	PVDF & PHBV	2	~25	24.75	~1150	/	6	[15]
	CS	Silicone rubber & Polyester	3	/	25	540	~60	8.92	[16]

**CS:** contact-separation, **SE:** single-electrode, **FT:** freestanding triboelectric-layer, **PDMS:** polydimethylsiloxane,
**PET:** polyethylene terephthalate, **PTFE:** poly tetra fluoroethylene, **Ni:** nickel, **EVOH:** ethylene vinyl alcohol
copolymer, **PHBV:** poly(3-hydroxybutyrate-co-3-hydroxyvalerate)

● **Our revision to the supplementary materials:**

**(i)** We added the explanation for Table S1 as Note S9: “We systematically sorted out the electrical
performance output and measurement conditions of the energy textiles to the best our knowledge.
These triboelectric textiles are mainly divided into three categories: All-yarn (breathable), All-fabric
(breathable) and Airtight (impermeable) as shown in Table S1. It should be noted that the voltages
and currents listed in table are the corresponding outputs of peak power density.” Corresponding

changes have been marked in red in the revised supplementary materials.

**(ii)** Table R1 was added as Table S1.

4. Authors should show the current of the triboelectric signal for their electronic textiles as well.

**Response:** Thank you for your kind advice. The current of the triboelectric signal for the e-textiles
has been shown in Figure 1e, Figure S1f, Figure S18a and Figure S18d of manuscript and
supplementary information. In order to make the experimental data more rigorous, we supplemented
the current signal corresponding to each voltage signal in the manuscript.

● **Effect of primary polarization direction on the voltage (under 100 MΩ) and short-current**
**signals of e-textiles.**

**Figure R7 (a), and (b)** Effect of the primary polarization direction of P(VDF-TrFE) nanofibers
ferroelectricity on the voltage (under 100 MΩ load) and short-circuit current of e-textiles.

● **Effect of ferroelectric layer thickness on the voltage (under 100 MΩ) and short-current**
**signals of e-textiles.**

 **Figure R8 (a), and (b)** Effect of inner ferroelectric thickness (D_{DFP-i} (When $D_{DFP-o} = 87 \mu\text{m}$)) on voltage (under
 $100\text{M}\Omega$ load) and short-circuit current in DFP e-textile. (c), and (d) Effect of outer ferroelectric thickness (D_{DFP-o}
 (When $D_{DFP-i} = 27 \mu\text{m}$)) on voltage (under $100 \text{M}\Omega$ load) and short-circuit current in DFP e-textile. (e), and (f) Effect
 of ferroelectric thickness (D_{SFP}) on voltage (under $100 \text{M}\Omega$ load) and short-circuit current in SFP e-textile.

 ● **Output voltage (under $100 \text{M}\Omega$) and short-current signals of e-textiles after repeated**
 **washing.**

**Figure R9 (a), and (b)** Output voltage (under 100 MΩ) and short-circuit current signals of the e-textile after repeated
 washing.

● **Our revision to the supplementary materials:**

(i) Figure R7 was added as Figure S6.

(ii) Figure R8a, b, c, and d were added as Figure S7. Figure R8e, and f were added to Figure S5.

(iii) Figure R9a and b were added as Figure S18c.

Corresponding changes have been marked in red in the revised supplementary materials.

**Thank you again for your valuable comments and suggestions.**

**Response to Reviewer #2**

**General comment:** *The author realized all-fiber tribo-ferroelectric synergistic e-texture with*
*thermal-moisture stability and comfortability. They discussed the tribo-ferroelectric synergistic*
*mechanism in mechanical-to-electrical energy conversion behavior and moisture wicking mechanism*
*of hierarchical nanofiber network in sweating state. Using ferroelectric polymer nanofibers showed*
*peak power density of 5.2 W/m², and breathability increased by 7 times. The manuscript was written*
*systematically and logically. However, it needs to improve some of points. I would like to recommend*
*the manuscript to be published at Nature Communication after the revision.*

**Response:** Thank you for the positive feedback. We have done our best to address the questions raised.
The replies to each of your concern are listed below.

1) *In Figure 1e and 1f, authors should confirm the performance of the device without inner*
*ferroelectric layer experimentally, even though they analyzed this by simulation.*

**Response:** Thanks for your valuable suggestions. As the reviewers said, the results of without inner
ferroelectric layer are not explicitly mentioned in the original manuscript, but this is undoubtedly a
very important part. Therefore, we have conducted additional experiments on the electrical output of
single-polarized ferroelectric (SFP) e-textile (without inner ferroelectric layer).

● **The tribo-ferroelectric synergistic model and electrical output performance of SFP e-**
**textiles.**

The effect of primary polarization direction (θ) and thickness (D_{SFP}) of ferroelectricity on the
performance of SFP e-textile were studied. When the detection between primary polarization and
effective electric field ($E_D + E_p$) change from same to opposite, the output voltage and current are
significantly improved (Figure R1b). Because the smaller the θ , the more difficult it is for dipoles to
reach the equilibrium position, resulting in a smaller residual polarization and lower performance.
The relationship between output performance and thickness is shown in Figure R1c, and d. The results
can be explained by the fact that the increasing of thickness leads to an opposite trend between the
amounts of dipoles and internal electric field intensity.

**Figure R1. (a)** Schematic diagram of tribo-ferroelectric synergistic model in single ferroelectric polarization (SFP)
 e-textile. **(b)** Influence of primary polarization direction (θ) of P(VDF-TrFE) nanofibers on the performance of SFP
 e-textile. **(c)**, and **(d)** Effect of ferroelectric layer thickness (D_{SFP}) on voltage (under 100 M Ω load) and short-circuit
 current of SFP e-textile.

**● Our revision to the manuscript:**

(i) We added “in dual-ferroelectric polarized (DFP) e-textile” (Please see page 5) and “Similar model
 for single ferroelectric polarized (SFP) e-textile is illustrated in Supplementary Fig. 5.” (Please see
 page 6) in the revised manuscript.

Corresponding changes have been marked in red in the revised manuscript.

(ii) We also added the polarization direction and thickness data (Red wireframe in Figure R2b, c) of
 the ferroelectric layer in SFP e-textile in Fig. 1e and f. We adjusted the subscript ($D_o \rightarrow D_{DFP-o}$ and D_i
 $\rightarrow D_{DFP-i}$) of the thickness of ferroelectric layer (Red wireframe in Figure R2 a) in Fig. 1d.

**Figure R2. Mark of the Corresponding changes in original manuscript Figure 1. (a)** Figure R2a corresponds
to Figure 1d in the original manuscript. **(b)** Figure R2b corresponds to Figure 1e in the original manuscript. **(c)**
Figure R2c corresponds to Figure 1f in the revised manuscript.

**• Our revision to the supplementary materials:**

**(i)** We added the discussion of tribo-ferroelectric synergistic enhancement model in SFP e-textile as
Note S1: “When the detection between primary polarization and effective electric field ($E_D + E_p$)
change from same to opposite, the output voltage and current are significantly improved (Figure S5b).
Because the smaller the θ , the more difficult it is for dipoles to reach the equilibrium position,
resulting in a smaller residual polarization and lower performance. The relationship between output
performance and thickness is shown in Figure S5c, and d. The results can be explained by the fact
that the increasing of thickness leads to an opposite trend between the amounts of dipoles and internal
electric field intensity.”.

**(ii)** Figure R1 were added as Fig. S5.

2. They need to mention the device size what they used in Figure 1e. 4×6 cm.?

**Response:** Thanks for your careful comment. It is mentioned in the experimental section in the
original manuscript (*Please see page 19*) that the size of e-textiles tested in experiment were unified
to 4×6 cm. For better demonstration, we have taken a photo of the e-textiles and test equipment used
in experiment.

**Figure R3 (a)** Digital photograph of e-textiles during experimental testing. The part marked by red ruler is the
effective friction zone, and the size is 4×6 cm. **(b)** Test status of e-textiles in vertical contact mode.

● **Our revision to the manuscript:**

We added “The e-textile tested in experiment were uniformly sized to 4×6 cm.” and “The error bars
correspond to standard deviation caused by the measurement noise.” (*Please see page 4*) in the
revised manuscript.

Corresponding changes have been marked in red in the revised manuscript.

● **Our revision to the supplementary materials:**

Figure R3 was added as Figure S4.

*3. In this paper, tribo-ferroelectric synergistic mechanism is one of the key factors to explain the high*
*performance of this hybrid system. They need to explain the charge behavior when it contacts each*
*other. For example, the reasons why the amounts of the charge transfer are different between first*
*contact and second contact. They have drawn the schematic without any evidence experimental and*
*simulation.*

**Response:** Thank you for your professional comments. We have described in detail the effects of
tribo-ferroelectric synergistic mechanism on tribo-charges transfer behavior during contact-
separation. In addition, we also designed unpolarized (UP), single ferroelectric polarized (SFP) and
dual-ferroelectric polarized (DFP) e-textiles to compared the charge density, short-circuit current,
surface potential and voltage (under 100 MΩ load) during contact-separation, which further verify
the tribo-ferroelectric synergistic mechanism.

● Detailed description of the effects of tribo-ferroelectric synergistic mechanism on tribo-charges transfer behavior in DFP e-textile.

The e-textile consists of two nanofiber nonwovens P(VDF-TrFE) and PA6 with opposite tribo-polarity for contact electrification, Ni-Cu fabric electrode for charge induction. The P(VDF-TrFE) nanofibers also act as a polymer ferroelectricity (Defined as inner/outer ferroelectric layers) for constructing tribo-ferroelectric synergistic enhancement effect. The in-situ polarization effect of electrospinning P(VDF-TrFE) is shown in Figure R4 i. When PA6 and P(VDF-TrFE) are in contact, they will acquire net opposite charges on their surfaces (ii). Once PA6 is separated from P(VDF-TrFE), electrons flow from fabric electrode II to fabric electrode I (iii). Meanwhile, the induced potential between two charged surfaces will result in a second polarization of P(VDF-TrFE) ferroelectricity. The polarization of P(VDF-TrFE) ferroelectricity will keep enhanced until the distance between two tribo-polarity materials reaches maximum (iv). As the separation distance decreases, the polarization of P(VDF-TrFE) ferroelectricity will gradually decrease until the PA6 contact with P(VDF-TrFE) again (v). Due to dielectric hysteresis, however, the polarization inside the inner and outer ferroelectric layers will not fully diminish, and the residual built-in dielectric polarization will act as positive and negative charge trap to enhance the capability of capturing charges during contact electrification (vi).

Figure R4. Working mechanism of the tribo-ferroelectric synergistic electric.

● Verifying the tribo-ferroelectric synergistic mechanism in e-textile.

Based on the “depolarization” idea of ferroelectricity by heat treatment, an unpolarized P(VDF-TrFE) nanofiber nonwovens was prepared without significantly changing its microstructure (Figure

1 R5). The process of electrospinning P (VDF-TrFE) depolarization is explained in detail in Figure S8
2 and Note S2.

**Figure R5. Depolarization of electrospun P(VDF-TrFE) nanofiber nonwovens.** (a) Schematic diagram of
depolarization of P(VDF-TrFE) ferroelectricity by heat treatment. (b) Micrographs of P(VDF-TrFE) nanofibers
before and after heat treatment. (c) As the temperature increases, piezoelectric coefficient (d_{33}) of P(VDF-TrFE)
ferroelectricity is continuously lowered to achieve the depolarization.

Figure R6a shows the process of charge accumulation in three e-textiles step by step. The UP e-
textile reaches a maximum value of $\sim 106 \mu\text{C}\cdot\text{m}^{-2}$ only after 1100 cycles operation, while the SFP and
DFP e-textiles increase with a much bigger slope and reach the maximum value of $\sim 161 \mu\text{C}\cdot\text{m}^{-2}$ after
2800 cycles and $\sim 320 \mu\text{C}\cdot\text{m}^{-2}$ after 6600 cycles. Similarly, with the continue of contact-separate
processes, the DFP e-textile has the fastest growth rate and the highest saturation current ($\sim 33 \mu\text{A}$).
The current growth rate and saturation current ($\sim 16 \mu\text{A}$) of SFP e-textile are higher than those of UP
e-textile ($\sim 10 \mu\text{A}$).

In addition, during the contact-separation process, we also tested the surface potential of tribo-
negative material (P(VDF-TrFE)). The surface potential was detected every 10 seconds, and the
distance between probe and the textile was fixed at 1.5 cm (Figure R6c). In 600 seconds, the surface
potential of P(VDF-TrFE) in UP e-textile increased from -0.3 kV to -1.05 kV, while the SFP and DFP
textiles increased from -0.34 kV and -0.29 kV to -1.91 kV and -3.1 kV, respectively. It visually shows
the accumulation of triboelectric charges step by step. And the charge accumulation rate and charge
density at the friction interface were significantly improved due to the tribo-ferroelectric synergistic
enhancement mechanism. As shown in Figure R6f, the output voltage results have similar trend.

In summary, we have compared the surface potential (surface charges), charge density (induction

charges), short-circuit current and voltage of UP, SFP and DFP e-textiles, the corresponding results
 strongly prove the proposed effect of ferroelectric polarization on surface charges transfer between
 opposite tribo-polarity polymer fibers.

 **Figure R6.** (a) The output charges density of UP, SFP and DFP e-textiles working continuously for 7200 cycles at
 a fixed frequency of 2.5 Hz. (b) The short-current of UP, SFP and DFP e-textiles working continuously for 7200
 cycles at a fixed frequency of 2.5 Hz. (c) Digital photo of the surface potential test device for e-textiles. (d)
 Schematic diagram of surface potential test procedure for UP, SFP and DFP e-textiles. (e) The surface potential
 (tribo-negative materials) versus time of UP, SFP and DFP e-textiles during contact and separation. (f) Comparison
 of output voltage (under $100\text{ M}\Omega$ load) and short-circuit current of UP, SFP and DFP e-textiles at a fixed frequency
 of 2.5 Hz.

● **Our revision to the manuscript:**

(i) We added “To better demonstrate the tribo-ferroelectric synergistic effect, we studied the
performance of e-textiles with or without ferroelectricity, i.e., DFP, SFP, and unpolarized (UP) e-
textiles. An UP e-textile can be obtained through depolarization of ferroelectric P(VDF-TrFE)
(Supplementary Fig. 8, and Supplementary Note 2). The charge density (Fig. 2a), short-circuit current
(Fig. 2b), surface potential (Fig. 2c) and output voltage (Supplementary Fig. 9 and Supplementary
Note 3) of UP, SFP and DFP e-textiles were tested and compared, respectively. The charge
accumulation rate and surface charge density of DFP e-textile are significantly higher than those of
SFP and UP e-textiles.” (Please see page 8) in the revised manuscript.

We added in the legend to Figure 1d “ E_{F1} and E_{F2} represent the Fermi level (gray dashed line) of
PA6 and P(VDF-TrFE) before contact, respectively. $E_{F1'}$ and $E_{F2'}$ represent the Fermi level (red
straight line) of PA6 and P(VDF-TrFE) after contact, respectively.” (Please see page 8) in the revised
manuscript.

We added “When tribo-polarity materials contact again, the residual polarization of ferroelectricity
(Fig. 1c, and d State 3) will enhance the capability of capturing charges therefore extra charges will
transfer after previous ones (Fig. 2e Next contact state). The Fermi level at the friction interface
remains equal, thus completing a cycle.” and “Detailed description about charge transfer behavior of
tribo-ferroelectric synergistic mechanism is shown in Supplementary Fig. 10 and Supplementary
Note 4.” (Please see page 9) in the revised manuscript.

We also added “The Keithley 2657A and Keithley 6514 were used to test the electrical output
performance of the e-textile. The surface potentials of the e-textiles were determined using an
electrostatic voltmeter (TREK 542A-2, USA), at a relative humidity of approximately 30 %
(Supplementary Fig. 9c).” (Please see page 20) in experimental section of the revised manuscript.

Corresponding changes have been marked in red in the revised manuscript.

(ii) We added the comparison data of charge density, short-circuit current and surface potential of UP,
SFP and DFP e-textiles during contact and separation as Figure 2 a, b, and c. (Please see page 7)

**Figure R7.** Comparison data of charge density, short-circuit current and surface potential during contact
separation of UP, SFP and DFP e-textiles were added as Figure 2 a, b, and c.

(iii) We corrected the expressions of “First contact state” and “Second contact state” to “Contact state”

and “Next contact state”, respectively. And We modified the polarization of the ferroelectric layer in
 “Separated state” and “Next contact state”. The modified part has been marked with red dotted frames,
 as shown in Figure R8.

 **Figure R8** Mark of the Corresponding changes in original manuscript Figure 2e.

**● Our revision to the supplementary materials:**

(i) We added detailed description of the effects of tribo-ferroelectric synergistic mechanism on tribo-
 charges transfer behavior in DFP e-textile as Note S4: “The e-textile consists of two nanofiber
 nonwovens P(VDF-TrFE) and PA6 with opposite tribo-polarity for contact electrification, Ni-Cu
 fabric electrode for charge induction. The P(VDF-TrFE) nanofibers also act as a polymer
 ferroelectricity (Defined as inner/outer ferroelectric layers) for constructing tribo-ferroelectric
 synergistic enhancement effect. The in-situ polarization effect of electrospinning P(VDF-TrFE) is
 shown in Figure S10 i. When PA6 and P(VDF-TrFE) are in contact, they will acquire net opposite
 charges on their surfaces (ii). Once PA6 is separated from P(VDF-TrFE), electrons flow from fabric
 electrode II to fabric electrode I (iii). Meanwhile, the induced potential between two charged surfaces
 will result in a second polarization of P(VDF-TrFE) ferroelectricity. The polarization of P(VDF-TrFE)
 ferroelectricity will keep enhanced until the distance between two tribo-polarity materials reaches
 maximum (iv). As the separation distance decreases, the polarization of P(VDF-TrFE) ferroelectricity
 will gradually decrease until the PA6 contact with P(VDF-TrFE) again (v). Due to dielectric hysteresis,
 however, the polarization inside the inner and outer ferroelectric layers will not fully diminish, and
 the residual built-in dielectric polarization will act as positive and negative charge trap to enhance the

capability of capturing charges during contact electrification (vi).”.

We added the design idea and preparation method of “depolarization” for electrospun nanofiber
nonwovens as Note S2: “In order to prepare an unpolarized P(VDF-TrFE) nanofiber nonwovens that
has the same microstructure as the originally polarized nanofiber ferroelectricity, we proposed the
idea of “**depolarization**” (Figure S8a). Here, we treated the electrospun P(VDF-TrFE) nanofiber
nonwovens at different temperatures to study the depolarization behavior and microstructure changes
(Figure S8b, c, and d). As the heat treatment temperature increases, the d_{33} of P(VDF-TrFE)
ferroelectricity decreases continuously. When temperature reaches 190 °C, the nanofibers begin to
melt and the microscopic morphology changes, which would change the surface friction behavior and
affect the charge transfer process. When the temperature is higher than 180 °C, the d_{33} value does not
decrease significantly, which indicates that the heat treatment at 180 °C for 3 h is a suitable
depolarization process of P(VDF-TrFE) without significantly changing its microstructure. Figure S8e
shows the circuit diagram of the d_{33} test. Therefore, the P(VDF-TrFE) obtained by heat treatment at
180 °C for 3 hours can be considered to have completely depolarized. This sample was used in the
preparation of unpolarized ferroelectric (UP) e-textile for control experiments.”.

We also added the comparison of electrical output performance of UP, SFP and DFP e-textiles as
Note S3: “Figure S9a shows the process of charge accumulation in three e-textiles step by step. The
UP e-textile reaches a maximum value of $\sim 106 \mu\text{C}\cdot\text{m}^{-2}$ only after 1100 cycles operation, while the
SFP and DFP e-textiles increase with a much bigger slope and reach the maximum value of ~ 161
$\mu\text{C}\cdot\text{m}^{-2}$ after 2800 cycles and $\sim 320 \mu\text{C}\cdot\text{m}^{-2}$ after 6600 cycles. Similarly, with the continue of contact-
separate processes, the DFP e-textile has the fastest growth rate and the highest saturation current
($\sim 33 \mu\text{A}$). The current growth rate and saturation current ($\sim 16 \mu\text{A}$) of SFP e-textile are higher than
those of UP e-textile ($\sim 10 \mu\text{A}$). In addition, during the contact-separation process, we also tested the
surface potential of tribo-negative material (P(VDF-TrFE)). It was detected every 10 seconds, and
the distance between probe and the textile was fixed at 1.5 cm (Figure S9c). In 600 seconds, the
surface potential of P(VDF-TrFE) in UP e-textile increased from -0.3 kV to -1.05 kV, while the SFP
and DFP textiles increased from -0.34 kV and -0.29 kV to -1.91 kV and -3.1 kV, respectively. It
visually shows the accumulation of triboelectric charges step by step. And the surface charge
accumulation rate and charge density at the friction interface were significantly improved due to the
tribo-ferroelectric synergistic enhancement mechanism. As shown in Figure S9d, the output voltage
results have similar trend.”.

(ii) Figure R4 was added to Figure S10. Figure R5 was modified and added to S8. Figure R6a (i, iii),
b (i, iii), c (i, ii), d, and f were added as Figure S9.

4. In Figure 3b and 3c, the PA6 and PAN have similar surface polarities measured by contact angle. And, PAN has 10 times larger pore size. It is necessary to explain why the PA6-PAN can evaporate the wafer more effectively, even though the PA6-PAN fabric has much thicker and much smaller pore size than the PAN fabric.

Response: We appreciate your comment, which inspires us to do more rethinking to our results. We have further explained the effect of pore size on water permeation and spreading processes, and added the discussion of PA6 nanofiber layer thickness on water evaporation rate.

● **The effect of pore size of fiber layer on water permeation and spreading processes.**

When sweat is in contact with porous PAN and PA6 fabric layers, the sweat infiltrates the hydrophilic nanofibers, and capillary force (Laplace pressure) of fiber channel can be expressed as [5-7]

$$P = \frac{4\gamma\cos\theta}{D} \quad (1)$$

where θ is the contact angle between water and the fabric, D is the pore diameter composed of nanofibers, γ is the surface tension of water in air. According to Equation (1), the Laplace pressure is inversely proportional to D . For PAN and PA6 fabrics, the capillary force (Figure R9 b) of the channel can be express as

$$P_{PAN} = \frac{4\gamma\cos\theta_{PAN}}{D_{PAN}},$$
$$P_{PA6} = \frac{4\gamma\cos\theta_{PA6}}{D_{PA6}} \quad (2)$$

Since the fabricated PAN and PA6 fibers have similar contact angles and the pore size of PAN is about 10 times larger than PA6, the sweat **permeation and spreading driving force** on PA6 fibers is theoretically faster than that of PAN.

$$\Delta P = P_{PA6} - P_{PAN} = \frac{4\gamma\cos\theta_{PA6}}{D_{PA6}} - \frac{4\gamma\cos\theta_{PAN}}{D_{PAN}} > 0 \quad (3)$$

And we have compared of water evaporation rates of cotton-PAN and cotton-PA6 fabrics at the same thickness (Figure R9 c). The experimental results further prove our discussion.

● **Evaluate the effect of PA6 layer thickness on water evaporation rate of the fabricated textiles.**

To evaluate the effect of PA6 layer thickness on water evaporation rate of the fabricated textiles, top layer (PAN) thickness and the volume of sweat (200 μ L) were kept constant. The increase in the thickness of PA6 layer enriches the mass of the layer, thereby enhancing the water absorption capacity of moisture wicking fabric and also giving rise to the wettability gradient between two layers. (From light blue to dark blue) [9], [10]. Since the PA6 layer has a faster penetration and spreading driving force (Figure R9b and c) than PAN layer, more sweat enrichment in PA6 layer will be more conducive

to evaporation. As shown in Figure R9d and e, with the increase of PA6 layer thickness (From 0 μm ,
 $40\pm 6 \mu\text{m}$, $70\pm 8 \mu\text{m}$ to $85\pm 5 \mu\text{m}$), the water evaporation rate of moisture wicking fabric increases.
 Since the thickness of $85\pm 5 \mu\text{m}$ of PA6 layer is enough to pull out almost all water from the top (PAN)
 layer, the water evaporation rate gradually reaches saturation as the thickness of PA6 further increases
 (from $85\pm 5 \mu\text{m}$, $95\pm 9 \mu\text{m}$ to $100\pm 6 \mu\text{m}$).

 **Figure R9. (a)** Cross-sectional SEM images of moisture wicking fabric with trilayered architecture. **(b)** Schematic
 illustration explaining the superposed capillary pressure difference (ΔP) in the composite porous membranes to
 drive the water penetration and wide spreading, which is found as a function of pore size. **(c)** Comparison of water
 evaporation rates of cotton-PAN and cotton-PA6 fabrics at the same thickness. **(d)** Schematic diagram of water
 (liquid water: black dotted line, and water vapor: blue dotted line) transport at different PA6 layer thicknesses. **(e)**
 Evaluate the effect of PA6 layer thickness on water evaporation rate of the fabricated textiles. PAN layer thickness
 was kept constant. The thickness of each fiber layer was controlled by the electrospinning time.

 ● **Our revision to the manuscript:**

**(i)** We added in the legend to Figure3a(iii): “Liquid water transport direction: black dotted line. Water
 vapor transport direction: blue dotted line. Wettability gradient: the transition from light blue to dark
 blue indicates that the water content in the fibrous layer changes from less to more.” (Please see page
 10) in revised manuscript.

We added “Compared with PAN microfibers, PA6 nanofibers have a higher sweat penetration and
spreading driving force as well as larger specific surface area, therefore have faster water evaporation
rate (Supplementary Fig. 12b). Furthermore, introduction of PA6 fibers layer can enhance the water
uptake capacity of moisture wicking fabric, and also give rise to the wettability gradient between
PAN and PA6 layer (34, 35). Thus more sweat enrichment in PA6 layer will be more conducive to
evaporation (Supplementary Fig. 12c, d and Supplementary Note 5).” and the expression “PAN and
PA6 nanofibers have different pore sizes” was modified to “the pore size of PAN is about 10 times
larger than that of PA6” (Please see page 11) in revised manuscript.

We modified the description of Figure 3d as “It shows that Laplace pressure facilitates the
penetration and spreading of water in cotton-PA6-PAN fabrics. The wicking effect is expected to be
helpful for creating comfortable microenvironments to skin.” (Please see page 11) in revised
manuscript.

Corresponding changes have been marked in red in revised manuscript.

(ii) We also added the wettability gradient (From light blue to dark blue) between PAN and PA6 layers
(Red wireframe in Figure R10) of moisture wicking fabric in Fig. 3a iii.

**Figure R10 Mark of the Corresponding changes in original manuscript Figure 3a iii.**

● **Our revision to the supplementary materials:**

(i) We added the effect of fiber pore size and PA6 fiber layer thickness on water evaporation rate as
Note S6: “To evaluate the effect of PA6 layer thickness on water evaporation rate of the fabricated
textiles, top layer (PAN) thickness and the volume of sweat (200 μ L) were kept constant. The increase
in the thickness of PA6 layer enriches the mass of the layer, thereby enhancing the water absorption
capacity of moisture wicking fabric and also giving rise to the wettability gradient between two layers.
(From light blue to dark blue) [3], [4]. Since the PA6 layer has a faster penetration and spreading
driving force (Figure S12b) than PAN layer, more sweat enrichment in PA6 layer will be more
conducive to evaporation. As shown in Figure S12c and d, with the increase of PA6 layer thickness
(From 0 μ m, 40 ± 6 μ m, 70 ± 8 μ m to 85 ± 5 μ m), the water evaporation rate of moisture wicking fabric
increases. Since the thickness of 85 ± 5 μ m of PA6 layer is enough to pull out almost all water from

the top (PAN) layer, the water evaporation rate gradually reaches saturation as the thickness of PA6
further increases (from $85\pm 5\ \mu\text{m}$, $95\pm 9\ \mu\text{m}$ to $100\pm 6\ \mu\text{m}$).”.

(ii) Figure R9 was modified and added to Figure S12.

*5 In Figure 4d, the device performance should be supported by cyclic data over time. How many*
*times is the data? Over time, the system should monitor the durability of the device by moisture*
*whether or not it has moisture wicking fabric. For example, even if there is no moisture wicking fabric*
*in the device, it is necessary to compare whether the device performance can be restored to its original*
*state or continuously deteriorated as the water evaporates.*

**Response:** We greatly appreciate the constructive suggestions and fully agree with the reviewer.
Figure 4d shows the effect of different amounts of sweat on the electrical properties of the fabric
during moisture wicking. The output voltage (Under load of $100\ \text{M}\Omega$) was tested after it has dropped
to a stable state. The voltage was read every 2 minutes, then the average and error were calculated
(All data was tested repeatedly 3 times under same conditions.). In addition, we have added cyclic
test of electrical output and surface relative humidity of e-textiles during multiple moisture wicking,
and compared the difference between the case with and without moisture wicking fabric, as shown in
Figure R11.

● **Cyclic data of electrical output and surface relative humidity of e-textiles during multiple**
**moisture wicking.**

As shown in Figure R11a(i), after introducing $210\ \text{g}\cdot\text{m}^{-2}$ of sweat onto the surface of hogskin, the
relative humidity of friction material gradually increased from 30 to $\sim 50\%$, and lasted for about 10
26 min at 50% RH. Correspondingly, the output voltage (under $100\ \text{M}\Omega$ load) of the e-textile was
27 gradually reduced from 1110 to $\sim 800\ \text{V}$, and maintained for about 10 min (The voltage data in Figure
4d is obtained at this stage). Subsequently, the relative humidity of friction material gradually
decreased to 30% and reached equilibrium, and the corresponding voltage also raised to equilibrium.
So far, the moisture wicking process have undergone a cycle, lasting about 30 minutes. For e-textile
without moisture wicking fabric, the relative humidity of friction surface raised from 30% to 83% ,
and maintained for about 60 minutes at a high relative humidity. The voltage also had a similar trend
at this stage, from 1100 to $\sim 300\ \text{V}$. The whole cycle time is about 160 minutes as shown in Figure
R11b(i).

At the end of each moisture wicking process, the output voltage of the e-textile can be gradually
recovered, but less than the initial output voltage. This may be caused by the influence of residual

salt such as sodium chloride (NaCl) in e-textile on output voltage. It is noteworthy that after washing
 and drying, the output voltage of the e-textile can return to the initial state.

 **Figure R11 Cyclic data of output voltage (Under 100 MΩ load) and surface humidity of e-textiles during**
 **multiple moisture wicking. (a) With moisture wicking fabric, (b) Without moisture wicking fabric.**

 ● **Detailed explanation for the effect of moisture wicking process on electrical output of e-**
 **textiles.**

First, for moisture wicking fabrics, the hydrophilic PAN-PA6 hierarchical micro/nano fibers network

has a high specific surface area, and sweat can rapidly penetrate, spread, and evaporate. The water
evaporation rate is much higher than the e-textile without PAN-PA6 layer, as shown in Figure 3c and
Figs. R11a, and b. Second, the hydrophobic and breathable cotton fabric acts as the innermost layer
of moisture wicking fabric, which can effectively prevent the liquid water in PA6 layer from
penetrating into fabric electrode. In addition, due to the barrier effect of hydrophobic cotton fabric, a
large amount of moisture will preferentially evaporate on both sides and inner surfaces of the moisture
wicking fabric, then only a small portion of water vapor will sequentially penetrate three hydrophobic
layers of cotton fabric ($CA \approx 138^\circ$), fabric electrode ($CA \approx 113^\circ$) and P(VDF-TrFE) ($CA \approx 135^\circ$) to the
interface of dielectric material. Therefore, the relative humidity on the surface of dielectric material
can be maintained at a low level (30–50 %) and the output voltage of corresponding e-textile is not
significantly reduced.

In the absence of moisture wicking fabric (Fig. R11b), since the specific surface area of
hydrophobic fabric electrode (contact angle is about 138°) is significantly smaller than PAN-PA6
micro/nano fiber, sweat is difficult to spread and diffuse on it and the water evaporation rate is very
low (Fig. 3c). When the human body is in a sweating state, a large amount of sweat will directly
contact with the fabric electrode and continuously penetrate into the surface of dielectric material.
And the whole e-textile is in a sweat-soaked state (Figure R11b ii and iii). This causes a significant
increase in relative humidity (30–83%) and conductivity (NaCl in sweat) at the friction interface
which leads to a significant decrease in contact electrification effect and a large loss of triboelectric
charges [2].

In summary, each layer of moisture wicking fabric performs its specific functions ultimately
achieving continuous, directional, rapid transport of moisture. In the case of introducing a sweat
amount of $210 \text{ g} \cdot \text{m}^{-2}$, the water evaporation rate (more than 3 times) and electrical properties (more
than 2.5 times) were significantly increased as compared with the e-textile having no moisture
wicking layer.

● **Our revision to the manuscript:**

(i) We added “Due to the barrier effect of hydrophobic cotton fabric (Fig. 3a iii), a large amount of
moisture will preferentially evaporate on both sides and inner surface of the moisture wicking fabric,
then only a small portion of water vapor will sequentially penetrate three hydrophobic layers of cotton
fabric ($CA \approx 138^\circ$), fabric electrode ($CA \approx 113^\circ$) and P(VDF-TrFE) ($CA \approx 135^\circ$) to the surface of
dielectric material. Therefore, the relative humidity on the dielectric material can maintain at a low
level (30–50 %) and the output voltage (under 100 M Ω) of corresponding e-textile will not be
significantly reduced (Supplementary Fig. 16a and Supplementary Note 7).” (Please see page 13 and
14) in the revised manuscript.

We added “This causes a significant increase in relative humidity (30–83%) and conductivity (due

to considerable amount of sodium chloride (NaCl) in sweat) (30) at the friction interface, leading to
a significant decrease in contact electrification effect and a large loss of triboelectric charges. In
addition, after sweat wicking circles, the output voltage of the e-textile can be basically recovered,
but less than the initial output voltage (Supplementary Fig. 16b and Supplementary Note 7). This is
due to the influence of residual salt such as NaCl in e-textile. It is noteworthy that after washing and
drying, the output voltage of the e-textile can return to the initial value.” (Please see page 14) in the
revised manuscript.

We also added “Relative humidity was tested by humidity measuring instrument (GM620,
Shanghai Tianzhi Intelligent Technology Co., Ltd., China).” (Please see page 20) in experimental
section of the revised manuscript.

Corresponding changes have been marked in red in revised manuscript.

(ii) We also added the direction of water vapor transport (Red wireframe in Figure R12) of moisture
wicking fabric in Figure 3a iii.

**Figure R12.** Mark of the Corresponding changes in original manuscript Figure 3a iii.

● **Our revision to the supplementary materials:**

(i) We added the cyclic test of electrical output and surface relative humidity of e-textiles during
multiple moisture wicking as Note S8: “As shown in Figure S16a(i), after introducing $210 \text{ g}\cdot\text{m}^{-2}$ of
sweat onto the surface of hogskin, the relative humidity of friction material gradually increased from
30 to $\sim 50\%$, and lasted for about 10 min at 50 % RH. Correspondingly, the output voltage (under
$100 \text{ M}\Omega$ load) of e-textile was gradually reduced from 1110 to $\sim 800 \text{ V}$, and maintained for about 10
24 min (The voltage data in Figure 4d is obtained at this stage). Subsequently, the relative humidity of
25 friction material gradually decreased to 30 % and reached equilibrium, and the corresponding voltage
also raised to equilibrium. So far, the moisture wicking process have undergone a cycle, lasting about
30 minutes. For e-textile without moisture wicking fabric, the relative humidity of friction surface
raised from 30 to 83%, and maintained for about 60 minutes at a high relative humidity. The voltage
also had a similar trend at this stage, from 1100 to $\sim 300 \text{ V}$. The whole cycle time is about 160 minutes
as shown in Figure S16b(i). At the end of each moisture wicking process, the output voltage of the
electronic textile can be gradually recovered, but less than the initial output voltage. This may be

caused by the influence of residual salt such as sodium chloride (NaCl) in e-textile on output voltage.
It is noteworthy that after washing and drying, the output voltage of e-textile can return to initial
state.”.

**(ii)** Figure R11 was added as Figure S16.

**Thank you again for your valuable comments and suggestions.**

**Response to Reviewer #3**

**General comment:** *The manuscript " All-fiber tribo-ferroelectric synergistic electronics with high*
*thermal-moisture stability and comfortability " suggested the use of tribo-ferroelectric fiber to*
*improve triboelectric output performance and good thermal-moisture stability of electronic clothing.*
*Using ferroelectric polymer nanofibers showed peak power density of 5.2 W/m², and breathability*
*increased by 7 times. Besides, material properties and specific functions of each layer are well*
*described, and various examples for applications using their electronic clothing are suggested. An*
*interesting manuscript could be published. The reviewer has a few minor comments as below.*

**Response:** Thank you for the positive feedback. We have carefully revised the manuscript according
to your comments. The replies to each of your concern are listed below.

*1. Need to describe the correct mechanism for triboelectric effect of tribo-ferroelectric synergistic*
*electronics in Fig. 2. As considering working principle of TENG, the output performance is*
*determined by the extent of the exchange of charges. However, there is no experimental result to prove*
*working mechanism that triboelectric performance change step by step.*

**Response:** Thank you very much for your professional advice on working mechanism. We have
described in detail the effects of tribo-ferroelectric synergistic mechanism on tribo-charges transfer
behavior during contact-separation. In addition, we also designed and conducted unpolarized (UP),
single ferroelectric polarized (SFP) and dual-ferroelectric polarized (DFP) e-textiles to compared the
charge density, short-circuit current, surface potential and voltage (under 100 MΩ load) during
contact and separation process, which further verify the tribo-ferroelectric synergistic mechanism.

● **Detailed description of the tribo-ferroelectric synergistic mechanism in e-textiles**

The e-textile consists of two nanofiber nonwovens P(VDF-TrFE) and PA6 with opposite tribo-
polarity for contact electrification, Ni-Cu fabric electrode for charge induction. The P(VDF-TrFE)
nanofibers also act as a polymer ferroelectricity (Defined as inner/outer ferroelectric layers) for
constructing tribo-ferroelectric synergistic enhancement effect. The in-situ polarization effect of
electrospinning P(VDF-TrFE) is shown in Figure R1 i. When PA6 and P(VDF-TrFE) are in contact,
they will acquire net opposite charges on their surfaces (ii). Once PA6 is separated from P(VDF-
TrFE), electrons flow from fabric electrode II to fabric electrode I (iii). Meanwhile, the induced
potential between two charged surfaces will result in a second polarization of P(VDF-TrFE)
ferroelectricity. The polarization of P(VDF-TrFE) ferroelectricity will keep enhanced until the

distance between two tribo-polarity materials reaches maximum (iv). As the separation distance
 decreases, the polarization of P(VDF-TrFE) ferroelectricity will gradually decrease until the PA6
 contact with P(VDF-TrFE) again (v). Due to dielectric hysteresis, however, the polarization inside
 the inner and outer ferroelectric layers will not fully diminish, and the residual built-in dielectric
 polarization will act as positive and negative charge trap to enhance the capability of capturing
 charges during contact electrification (vi).

 **Figure R1** Working mechanism of the tribo-ferroelectric synergistic electrics.

 ● **Verifying the tribo-ferroelectric synergistic mechanism in e-textiles.**

Based on the "**depolarization**" idea of ferroelectricity by heat treatment, an unpolarized P(VDF-TrFE)
 nanofiber nonwovens was prepared without significantly changing its microstructure (Figure R2).
 The process of electrospinning P (VDF-TrFE) depolarization is explained in detail in Figure S8 and
 Note S2.

**Figure R2. Depolarization of electrospun P(VDF-TrFE) nanofiber nonwovens.** (a) Schematic diagram of
depolarization of P(VDF-TrFE) ferroelectricity by heat treatment. (b) Micrographs of P(VDF-TrFE) nanofibers
before and after heat treatment. (c) As the temperature increases, piezoelectric coefficient (d_{33}) of P(VDF-TrFE)
ferroelectricity is continuously lowered to achieve the depolarization.

**Figure R3.** (a) The output charges density of UP, SFP and DFP e-textiles working continuously for 7200 cycles at
a fixed frequency of 2.5 Hz. (b) The short-current of UP, SFP and DFP e-textiles working continuously for 7200
cycles at a fixed frequency of 2.5 Hz. (c) Digital photo of the surface potential test device for e-textiles. (d)
Schematic diagram of surface potential test procedure for UP, SFP and DFP e-textiles. (e) The surface potential
(tribo-negative materials) versus time of UP, SFP and DFP e-textiles during contact and separation. (f) Comparison
of output voltage (under 100 M Ω load) and short-circuit current of UP, SFP and DFP e-textiles at a fixed frequency
of 2.5 Hz.

Figure R3a shows the process of charge accumulation in three e-textiles step by step. The UP e-
textile reaches a maximum value of $\sim 106 \mu\text{C}\cdot\text{m}^{-2}$ only after 1100 cycles operation, while the SFP and
DFP e-textiles increase with a much bigger slope and reach the maximum value of $\sim 161 \mu\text{C}\cdot\text{m}^{-2}$ after

2800 cycles and $\sim 320 \mu\text{C}\cdot\text{m}^{-2}$ after 6600 cycles. Similarly, with the continue of contact-separate
processes, the DFP e-textile has the fastest growth rate and the highest saturation current ($\sim 33 \mu\text{A}$).
The current growth rate and saturation current ($\sim 16 \mu\text{A}$) of SFP e-textile are higher than those of UP
e-textile ($\sim 10 \mu\text{A}$).

In addition, during the contact-separation process, we also tested the surface potential of tribo-
negative material (P(VDF-TrFE)). The surface potential was detected every 10 seconds, and the
distance between probe and the textile was fixed at 1.5 cm (Figure R3c). In 600 seconds, the surface
potential of P(VDF-TrFE) in UP e-textile increased from -0.3 kV to -1.05 kV, while the SFP and DFP
textiles increased from -0.34 kV and -0.29 kV to -1.91 kV and -3.1 kV, respectively. It visually shows
the accumulation of triboelectric charges step by step. And the charge accumulation rate and charge
density at the friction interface were significantly improved due to the tribo-ferroelectric synergistic
enhancement mechanism. As shown in Figure R3f, the output voltage results have similar trend.

In summary, we have compared the surface potential (surface charges), charge density (induction
charges), short-circuit current and voltage of UP, SFP and DFP e-textiles, the corresponding results
strongly prove the proposed effect of ferroelectric polarization on surface charges transfer between
opposite tribo-polarity polymer fibers.

● **Our revision to the manuscript:**

(i) We added “To better demonstrate the tribo-ferroelectric synergistic effect, we studied the
performance of e-textiles with or without ferroelectricity, i.e., DFP, SFP, and unpolarized (UP) e-
textiles. An UP e-textile can be obtained through depolarization of ferroelectric P(VDF-TrFE)
(Supplementary Fig. 8, and Supplementary Note 2). The charge density (Fig. 2a), short-circuit current
(Fig. 2b), surface potential (Fig. 2c) and output voltage (Supplementary Fig. 9 and Supplementary
Note 3) of UP, SFP and DFP e-textiles were tested and compared, respectively. The charge
accumulation rate and surface charge density of DFP e-textile are significantly higher than those of
SFP and UP e-textiles.” (Please see page 8) in the revised manuscript.

We added in the legend to Figure 1d “ E_{F1} and E_{F2} represent the Fermi level (gray dashed line) of
PA6 and P(VDF-TrFE) before contact, respectively. $E_{F1'}$ and $E_{F2'}$ represent the Fermi level (red
straight line) of PA6 and P(VDF-TrFE) after contact, respectively.” (Please see page 8) in the revised
manuscript.

We added “When tribo-polarity materials contact again, the residual polarization of
ferroelectricity (Fig. 1c, and d State 3) will enhance the capability of capturing charges therefore extra
charges will transfer after previous ones (Fig. 2e Next contact state). The Fermi level at the friction
interface remains equal, thus completing a cycle.” and “Detailed description about charge transfer
behavior of tribo-ferroelectric synergistic mechanism is shown in Supplementary Fig. 10 and
Supplementary Note 4.” (Please see page 9) in the revised manuscript.

We also added “The Keithley 2657A and Keithley 6514 were used to test the electrical output
 performance of the e-textile. The surface potentials of the e-textiles were determined using an
 electrostatic voltmeter (TREK 542A-2, USA), at a relative humidity of approximately 30 %
 (Supplementary Fig. 9c).” (Please see page 20) in experimental section of the revised manuscript.

Corresponding changes have been marked in red in the revised manuscript.

 (ii) We also added the comparison data of charge density, short-circuit current and surface potential
 of UP, SFP and DFP e-textiles during contact separation as Figure 2 a, b, and c. (Please see page 7)

 **Figure R4.** Comparison data of charge density, short-circuit current and surface potential during contact
 separation of UP, SFP and DFP e-textiles were added as Figure 2 a, b, and c.

 (iii) We corrected the expressions of “First contact state” and “Second contact state” to “Contact state”
 and “Next contact state”, respectively. And We modified the polarization of the ferroelectric layer in
 “Separated state” and “Next contact state”. The modified part has been marked with red dotted frames,
 as shown in Figure R5.

**Figure R5** Mark of the Corresponding changes in original manuscript Figure 2e.

● **Our revision to the supplementary materials:**

(i) We added detailed description of the effects of tribo-ferroelectric synergistic mechanism on tribo-
charges transfer behavior in DFP e-textile as Note S4: “The e-textile consists of two nanofiber
nonwovens P(VDF-TrFE) and PA6 with opposite tribo-polarity for contact electrification, Ni-Cu
fabric electrode for charge induction. The P(VDF-TrFE) nanofibers also act as a polymer
ferroelectricity (Defined as inner/outer ferroelectric layers) for constructing tribo-ferroelectric
synergistic enhancement effect. The in-situ polarization effect of electrospinning P(VDF-TrFE) is
shown in Figure S10 i. When PA6 and P(VDF-TrFE) are in contact, they will acquire net opposite
charges on their surfaces (ii). Once PA6 is separated from P(VDF-TrFE), electrons flow from fabric
electrode II to fabric electrode I (iii). Meanwhile, the induced potential between two charged surfaces
will result in a second polarization of P(VDF-TrFE) ferroelectricity. The polarization of P(VDF-TrFE)
ferroelectricity will keep enhanced until the distance between two tribo-polarity materials reaches
maximum (iv). As the separation distance decreases, the polarization of P(VDF-TrFE) ferroelectricity
will gradually decrease until the PA6 contact with P(VDF-TrFE) again (v). Due to dielectric hysteresis,
however, the polarization inside the inner and outer ferroelectric layers will not fully diminish, and
the residual built-in dielectric polarization will act as positive and negative charge trap to enhance the
capability of capturing charges during contact electrification (vi).”.

We added the design idea and preparation method of “depolarization” for electrospun nanofiber
nonwovens as Note S2: “It is well known that the electrospinning process has in situ poling effect
and therefore induces preferred dipole orientation in P(VDF-TrFE) nanofiber ferroelectricity [1]. In
order to prepare an unpolarized P(VDF-TrFE) nanofiber nonwoven that has the same microstructure
as the originally polarized nanofiber ferroelectricity, we proposed the idea of depolarization (Figure
S8a). The ideas and experimental results are discussed as follows. Here, we treated the electrospun
P(VDF-TrFE) nanofiber nonwovens at different temperatures to study the depolarization behavior
and microstructure changes (Figure S8b, c and d). As the heat treatment temperature increases, the
d_{33} of P(VDF-TrFE) ferroelectricity decreases continuously. When temperature reaches 190 °C, the
nanofibers begin to melt and the microscopic morphology changes, which would change the surface
friction behavior and affect the charge transfer process. When the temperature is higher than 180 °C,
the d_{33} value does not decrease significantly, which indicates that the heat treatment at 180 °C for 3 h
is a suitable depolarization process of P(VDF-TrFE) without significantly changing its microstructure.
Figure S8e shows the circuit diagram of the d_{33} test. Therefore, the P(VDF-TrFE) obtained by heat
treatment at 180 °C for 3 hours can be considered to have completely depolarized. This sample was
use in the preparation of unpolarized ferroelectric (UP) e-textile for control experiments.”.

We also added the discussion of electrical output performance of UP, SFP and DFP e-textiles as

**Note S3:** “Figure S9a shows the process of charge accumulation in three e-textiles step by step. The
UP e-textile reaches a maximum value of $\sim 106 \mu\text{C}\cdot\text{m}^{-2}$ only after 1100 cycles operation, while the
SFP and DFP e-textiles increase with a much bigger slope and reach the maximum value of ~ 161
$\mu\text{C}\cdot\text{m}^{-2}$ after 2800 cycles and $\sim 320 \mu\text{C}\cdot\text{m}^{-2}$ after 6600 cycles. Similarly, with the continue of contact-
separate processes, the DFP e-textile has the fastest growth rate and the highest saturation current
($\sim 33 \mu\text{A}$). The current growth rate and saturation current ($\sim 16 \mu\text{A}$) of SFP e-textile are higher than
those of UP e-textile ($\sim 10 \mu\text{A}$). In addition, during the contact-separation process, we also tested the
surface potential of tribo-negative material (P(VDF-TrFE)). It was detected every 10 seconds, and
the distance between probe and the textile was fixed at 1.5 cm (Figure S9c). In 600 seconds, the
surface potential of P(VDF-TrFE) in UP e-textile increased from -0.3 kV to -1.05 kV, while the SFP
and DFP textiles increased from -0.34 kV and -0.29 kV to -1.91 kV and -3.1 kV, respectively. It
visually shows the accumulation of triboelectric charges step by step. And the surface charge
accumulation rate and charge density at the friction interface were significantly improved due to the
tribo-ferroelectric synergistic enhancement mechanism. As shown in Figure S9d, the output voltage
results have similar trend.”.

**(ii)** Figure R1 was added to Figure S10. Figure R2 was modified and added to S8. Figure R3a (i, iii),
b (i, iii), c (i, ii), d, and f were added as Figure S9.

*2. Need more detailed descriptions about moisture-wicking effect for electronic fabric. How can sweat*
*be transpired out of a fabric without significantly reducing the triboelectric output voltage?*

**Response:** Thank you very much for your kind advice. We have further explained the effect of pore
size on water permeation and spreading processes, and added the discussion of PA6 nanofiber layer
thickness on water evaporation rate. In addition, we also compared the humidity of dielectric layer
surface (friction material P(VDF-TrFE)) and voltage (under $100 \text{ M}\Omega$ load) of e-textile with/without
moisture wicking fabric layer during sweat evaporation. The spreading and evaporation paths of sweat
in the process of moisture wicking was explained in detail.

**● The effect of pore size on water permeation and spreading processes.**
When sweat is in contact with porous PAN and PA6 fabric layers, the sweat infiltrates the hydrophilic
nanofibers, and capillary force (Laplace pressure) of fiber channel can be expressed as [5-7]

$$33 \quad P = \frac{4\gamma\cos\theta}{D} \quad (1)$$

where θ is the contact angle between water and the fabric, D is the pore diameter composed of
nanofibers, γ is the surface tension of water in air. According to Equation (1), the Laplace pressure is

1 inversely proportional to D . For PAN and PA6 fabrics, the capillary force (Figure R6 b) of the channel
 2 can be express as

$$3 \quad P_{PAN} = \frac{4\gamma\cos\theta_{PAN}}{D_{PAN}},$$

$$4 \quad P_{PA6} = \frac{4\gamma\cos\theta_{PA6}}{D_{PA6}} \quad (2)$$

Since the fabricated PAN and PA6 fibers have similar contact angles and the pore size of PAN is
 about 10 times larger than PA6, the sweat **permeation and spreading driving force** on PA6 fibers
 is theoretically faster than that of PAN.

$$8 \quad \Delta P = P_{PA6} - P_{PAN} = \frac{4\gamma\cos\theta_{PA6}}{D_{PA6}} - \frac{4\gamma\cos\theta_{PAN}}{D_{PAN}} > 0 \quad (3)$$

And we have compared of water evaporation rates of cotton-PAN and cotton-PA6 fabrics at the same
 thickness (Figure R6 c). The experimental results further prove our discussion.

**Figure R6. (a)** Cross-sectional SEM images of moisture wicking fabric with trilayered architecture. **(b)** Schematic
 illustration explaining the superposed capillary pressure difference (ΔP) in the composite porous membranes to
 drive the water penetration and wide spreading, which is found as a function of pore size. **(c)** Comparison of water
 evaporation rates of cotton-PAN and cotton-PA6 fabrics at the same thickness. **(d)** Schematic diagram of water
 (Liquid water: black dotted line, and water vapor: blue dotted line) transport at different PA6 layer thicknesses.
 Wettability gradient: the transition from light blue to dark blue indicates that the water content in the fibrous layer

changes from less to more. (e) Evaluate the effect of PA6 layer thickness on water evaporation rate of the fabricated
textiles. PAN layer thickness was kept constant. The thickness of each fiber layer was controlled by the
electrospinning time.

● **Evaluate the effect of PA6 layer thickness on water evaporation rate of the fabricated**
**textiles.**

To evaluate the effect of PA6 layer thickness on water evaporation rate of the fabricated textiles, top
layer (PAN) thickness and the volume of sweat (200 μL) were kept constant. The increase in the
thickness of PA6 layer enriches the mass of the layer, thereby enhancing the water absorption capacity
of moisture wicking fabric and also giving rise to the wettability gradient between two layers. (From
light blue to dark blue) [9], [10]. Since the PA6 layer has a faster penetration and spreading driving
force (Figure R6b and c) than PAN layer, more sweat enrichment in PA6 layer will be more conducive
to evaporation. As shown in Figure R6d and e, with the increase of PA6 layer thickness (From 0 μm ,
$40\pm 6 \mu\text{m}$, $70\pm 8 \mu\text{m}$ to $85\pm 5 \mu\text{m}$), the water evaporation rate of moisture wicking fabric increases.
Since the thickness of $85\pm 5 \mu\text{m}$ of PA6 layer is enough to pull out almost all water from the top (PAN)
layer, the water evaporation rate gradually reaches saturation as the thickness of PA6 further increases
(from $85\pm 5 \mu\text{m}$, $95\pm 9 \mu\text{m}$ to $100\pm 6 \mu\text{m}$).

● **Effect of moisture wicking process on electrical signals of e-textiles**

First, for moisture wicking fabrics, the hydrophilic PAN-PA6 hierarchical micro/nano fibers network
has a high specific surface area, and sweat can rapidly penetrate, spread, and evaporate. The water
evaporation rate is much higher than the e-textile without PAN-PA6 layer, as shown in Figure 3c and
Figs. R7a, and b. Second, the hydrophobic and breathable cotton fabric acts as the innermost layer of
moisture wicking fabric, which can effectively prevent the liquid water in PA6 layer from penetrating
into fabric electrode. In addition, due to the barrier effect of hydrophobic cotton fabric, a large amount
of moisture will preferentially evaporate on both sides and inner surface of the moisture wicking
fabric, and only a small portion of water vapor will sequentially penetrate three hydrophobic layers
of cotton fabric ($CA\approx 138^\circ$), fabric electrode ($CA\approx 113^\circ$) and P(VDF-TrFE) ($CA\approx 135^\circ$) to the
interface of dielectric material. Therefore, the relative humidity on the surface of dielectric material
can be maintained at a low level (30–50 %) and the output voltage of corresponding e-textile is not
significantly reduced.

In the absence of moisture wicking fabric (Fig. R7b), since the specific surface area of hydrophobic
fabric electrode (contact angle is about 138°) is significantly smaller than PAN-PA6 micro/nano fiber,
sweat is difficult to spread and diffuse on it and the water evaporation rate is very low (Fig.3c). When
the human body is in a sweating state, a large amount of sweat will directly contact with the fabric
electrode and continuously penetrate into the surface of dielectric material. And the whole e-textile is

in a sweat-soaked state (Figure R7b ii and iii). This causes a significant increase in relative humidity
 (30 to 83%) and conductivity (NaCl in sweat) at the friction interface which leads to a significant
 decrease in contact electrification effect and a large loss of triboelectric charges [2].
 In summary, each layer of moisture wicking fabric performs its specific functions ultimately
 achieving continuous, directional, rapid transport of moisture. In the case of introducing a sweat
 amount of $210 \text{ g}\cdot\text{m}^{-2}$, the water evaporation rate (more than 3 times) and electrical properties (more
 than 2.5 times) were significantly increased as compared with the e-textile having no moisture
 wicking layer.

 **Figure R7** The humidity of dielectric layer surface and voltage (under $100 \text{ M}\Omega$ load) of the e-textile (a) with/ (b)
 without moisture wicking fabric during sweat evaporation.

 ● **Our revision to the manuscript:**

(i) We added in the legend to Figure3a(iii): “Liquid water transport direction: black dotted line. Water
 vapor transport direction: blue dotted line. Wettability gradient: the transition from light blue to dark
 blue indicates that the water content in the fibrous layer changes from less to more.” (Please see page
 10) in revised manuscript.

We added “Compared with PAN microfibers, PA6 nanofibers have a higher sweat penetration and
spreading driving force as well as larger specific surface area, therefore have faster water evaporation
rate (Supplementary Fig. 12b). Furthermore, introduction of PA6 fibers layer can enhance the water
uptake capacity of moisture wicking fabric, and also give rise to the wettability gradient between
PAN and PA6 layer (34, 35). Thus more sweat enrichment in PA6 layer will be more conducive to
evaporation (Supplementary Fig. 12c, d and Supplementary Note 5).” and the expression “PAN and
PA6 nanofibers have different pore sizes” was modified to “the pore size of PAN is about 10 times
larger than that of PA6” (*Please see page 11*) in revised manuscript.

We modified the description of Figure 3d as “It shows that Laplace pressure facilitates the
penetration and spreading of water in cotton-PA6-PAN fabrics. The wicking effect is expected to be
helpful for creating comfortable microenvironments to skin.” (*Please see page 11*) in revised
manuscript.

We also added “Relative humidity was tested by humidity measuring instrument (GM620,
Shanghai Tianzhi Intelligent Technology Co., Ltd., China).” (*Please see page 20*) in experimental
section of the revised manuscript.

**(ii)** We added “Due to the barrier effect of hydrophobic cotton fabric (Fig. 3a iii), a large amount of
moisture will preferentially evaporate on both sides and inner surface of the moisture wicking fabric,
then only a small portion of water vapor will sequentially penetrate three hydrophobic layers of cotton
fabric ($CA \approx 138^\circ$), fabric electrode ($CA \approx 113^\circ$) and P(VDF-TrFE) ($CA \approx 135^\circ$) to the surface of
dielectric material. Therefore, the relative humidity on the dielectric material can maintain at a low
level (30–50 %) and the output voltage (under 100 M Ω) of corresponding e-textile will not be
significantly reduced (Supplementary Fig. 16a and Supplementary Note 7).” (*Please see page 13 and*
*14*) in the revised manuscript.

We also added “This causes a significant increase in relative humidity (30–83%) and conductivity
(due to considerable amount of sodium chloride (NaCl) in sweat) (30) at the friction interface, leading
to a significant decrease in contact electrification effect and a large loss of triboelectric charges.”
(*Please see page 14*) in the revised manuscript.

Corresponding changes have been marked in red in revised manuscript.

**(iii)** We added the wettability gradient (From light blue to dark blue) and the direction of water vapor
transport between PAN and PA6 layers (Red wireframe in Figure R8) of moisture wicking fabric in
Fig. 3a iii.

Figure R8. Mark of the Corresponding changes in original manuscript Figure 3a iii.

● **Our revision to the supplementary materials:**

(i) We added the effect of fiber pore size and PA6 fiber layer thickness on water evaporation rate as Note S6: “To evaluate the effect of PA6 layer thickness on water evaporation rate of the fabricated textiles, top layer (PAN) thickness and the volume of sweat (200 μL) were kept constant. The increase in the thickness of PA6 layer enriches the mass of the layer, thereby enhancing the water absorption capacity of moisture wicking fabric and also giving rise to the wettability gradient between two layers. (From light blue to dark blue) [3], [4]. Since the PA6 layer has a faster penetration and spreading driving force (Figure S12b) than PAN layer, more sweat enrichment in PA6 layer will be more conducive to evaporation. As shown in Figure S12c and d, with the increase of PA6 layer thickness (From 0 μm , $40\pm 6 \mu\text{m}$, $70\pm 8 \mu\text{m}$ to $85\pm 5 \mu\text{m}$), the water evaporation rate of moisture wicking fabric increases. Since the thickness of $85\pm 5 \mu\text{m}$ of PA6 layer is enough to pull out almost all water from the top (PAN) layer, the water evaporation rate gradually reaches saturation as the thickness of PA6 further increases (from $85\pm 5 \mu\text{m}$, $95\pm 9 \mu\text{m}$ to $100\pm 6 \mu\text{m}$).”. We also added the change electrical output and surface relative humidity of e-textiles during multiple moisture wicking as Note S8: “As shown in Figure S16a(i), after introducing $210 \text{ g}\cdot\text{m}^{-2}$ of sweat onto the surface of hogskin, the relative humidity of friction material gradually increased from 30 to $\sim 50 \%$, and lasted for about 10 min at 50 % RH. Correspondingly, the output voltage (under 100 $\text{M}\Omega$ load) of e-textile was gradually reduced from 1110 to $\sim 800 \text{ V}$, and maintained for about 10 min (The voltage data in Figure 4d is obtained at this stage). Subsequently, the relative humidity of friction material gradually decreased to 30 % and reached equilibrium, and the corresponding voltage also raised to equilibrium. So far, the moisture wicking process have undergone a cycle, lasting about 30 minutes. For e-textile without moisture wicking fabric, the relative humidity of friction surface raised from 30 to 83%, and maintained for about 60 minutes at a high relative humidity. The voltage also had a similar trend at this stage, from 1100 to $\sim 300 \text{ V}$. The whole cycle time is about 160 minutes as shown in Figure S16b(i). At the end of each moisture wicking process, the output voltage of the electronic textile can be gradually recovered, but less than the initial output voltage. This may be caused by the influence

of residual salt such as sodium chloride (NaCl) in e-textile on output voltage. It is noteworthy that
after washing and drying, the output voltage of e-textile can return to initial state.”.

(ii) Figure R5, and R6 was modified and added to Figure S12 and S16.

**Thank you again for your valuable comments and suggestions.**

Reference in Responses

- 1. Lei, T. et al. Electrospinning-induced preferred dipole orientation in PVDF fibers. *J. Mater. Sci.*
**50**, 4342-4347 (2015).
- 2. Nguyen, Vu., Zhu, R., & Yang, R. Environmental effects on nanogenerators. *Nano Energy* **14**,
49-61 (2015).
- 3. Chen, Q., Shen, Y., Zhang, S. & Zhang, Q. M. Polymer-based dielectrics with high energy storage
density. *Annu. Rev. Mater. Res.* **45**, 433-458 (2015).
- 4. Shen, Y., Zhang, X., Li, M., Lin, Y. & Nan, C. Polymer nanocomposite dielectrics for electrical
energy storage. *Natl. Sci. Rev.* **4**, 23-25 (2017).
- 5. Tagantsev, A. K., & Stolichnov, I. A. Injection-controlled size effect on switching of ferroelectric
thin films. *Appl. Phys. Lett.* **74**, 1326-1328 (1999).
- 6. Zhang, L. Y. et al. Dielectric Physics Ch1 (Xi'an Jiaotong University Publishing House, Xi'an,
1991).
- 7. Wang, X. et al. Biomimetic fibrous Murray membranes with ultrafast water transport and
evaporation for smart moisture-wicking fabrics. *ACS nano* **13**, 1060-1070 (2018).
- 8. Miao, D., Huang, Z., Wang, X., Yu, J. & Ding, B. Continuous, Spontaneous, and Directional
Water Transport in the Trilayered Fibrous Membranes for Functional Moisture Wicking Textiles.
*Small* **14**, 1801527 (2018).
- 9. Babar, A. A. et al. Breathable and colorful cellulose acetate-based Nanofibrous membranes for
directional moisture transport. *ACS Appl. Mater. Interface* **10**, 22866-22875 (2018).
- 10. Yoo, S. & Barker, R. Moisture management properties of heat-resistant workwear fabrics—
Effects of hydrophilic finishes and hygroscopic fiber blends. *Text. Res. J.* **74**, 995-1000 (2004).

REVIEWERS' COMMENTS:

Reviewer #1 (Remarks to the Author):

The authors have addressed all questions that reviewers concerned, I do not have further questions and recommend Nature Communication accept this manuscript.

Reviewer #2 (Remarks to the Author):

The authors clearly responded to the reviewer's comments. And, the modified manuscript is well organized. I recommend publication in Nature Communications for the paper in its present form.

Reviewer #3 (Remarks to the Author):

Well revised. I think it is ready to be accepted.